



# Monitoring snowpack outflow volumes and their isotopic composition to better understand streamflow generation during rain-on-snow events

Andrea Rücker[1,2], Stefan Boss[1], James W. Kirchner[2,1], Jana von Freyberg[2,1]

[1]Swiss Federal Institute for Forest, Snow and Landscape Research (WSL), Zürcherstrasse 111, 8903 Birmensdorf, Switzerland
[2] Department of Environmental System Science, ETH Zürich, Universitätsstrasse 16, 8092 Zurich, Switzerland

*Correspondence to*: Andrea Rücker (andrea.ruecker@wsl.ch), Jana von Freyberg (jana.vonfreyberg@usys.ethz.ch)

**Abstract.** Rain-on-snow (ROS) events in mountainous catchments can cause enhanced snowmelt, leading to destructive winter floods. However, due to differences in topography and vegetation cover, the generation of snowpack outflow and its contribution to streamflow is spatially and temporally variable during ROS events. In order to adequately predict such flood events with hydrological models, an enhanced process understanding of the contribution of rainwater and snowmelt to stream water is needed.

In this study, we monitored and sampled snowpack outflow with fully automated snowmelt lysimeter systems installed at three different locations in a pre-Alpine catchment in Central Switzerland. We measured snowpack outflow volumes during the winters of 2017 and 2018, as well as snowpack outflow isotopic compositions for winter 2017. Snowpack outflow volumes were highly variable in time and space reflecting differences in snow accumulation and melt. In winter 2017, around 815 mm snowpack outflow occurred at our reference site (grassland 1220 m above sea level, m asl), whereas snowpack outflow was 16 % less at the nearby forest site (1185 m asl), and 62 % greater at another grassland site located 200-meter higher (1420 m asl). A detailed analysis of ten ROS events showed that the snowpack outflow volumes could be explained mainly by rainfall volume and initial snow depth.

The isotope signal of snowpack outflow was more damped than that of incoming rainfall at all three sites, with the most damped signal at the high-elevation site because its snowpack was the thickest and residence times of liquid water in the snowpack were the longest, thus enhancing isotopic mixing in the snowpack. The contribution of snowpack outflow to streamflow, estimated by isotope-based two-component end member mixing analysis, differed substantially among the three lysimeter sites. Because the study catchment vegetation is a mixture of grassland and forest and altitudes range from 1000 to 1500 m asl, the catchment-average contribution of snowpack outflow to stream discharge is likely to lie between the end member mixing estimates derived from the three site-specific datasets. Thus, our hydrograph separation estimates based on the measurements from the three lysimeter sites provide a range of snowpack outflow contributions to discharge from





different parts of the study area. This information may be useful for improving hydrological models in snow-dominated catchments.

## 1 Introduction

Over the past 50 years, rain-on-snow (ROS) events have become more frequent in snow-dominated catchments, because
increasing global mean air temperature has led to greater fractions of winter precipitation falling as rain instead of snow (Barnett et al., 2005; Beniston and Stoffel, 2016; Hartmann et al., 2013; Stewart, 2009). In Switzerland, mean air temperature is predicted to increase by up to 1.6 °C by 2050 (Swiss Academics Reports, 2016), and thus the rain-snow transition zone is likely to expand to altitudes above 2000 m above sea level (asl), while altitudes below 1500 m asl might more frequently register rain on snow-free conditions (Beniston, 2003; McCabe et al., 2007; Surfleet and Tullos, 2013; Zierl
and Bugmann, 2005).

Rain on snow can either be retained in the snowpack or it can enhance the melting of the snowpack, so that the snowpack can either reduce or amplify the volume of water reaching the ground surface, relative to snow-free conditions (Kattelmann, 1987; Lee et al., 2010). In the past, some ROS events that caused enhanced snow melt have led to severe floods (e.g., Garvelmann et al., 2015; Kroczynski, 2004; MacDonald and Hoffman, 1995; Marks et al., 1998; Sui and Koehler, 2001;
Wever et al., 2014). Peak flows caused by ROS events result from a complex interplay of processes that mainly depend on the initial snowpack properties, rainfall characteristics and energy fluxes (Colbeck, 1977; Garvelmann et al., 2014; Würzer et al., 2016), as well as antecedent catchment wetness, and thus predictions of flood responses during ROS events can be highly uncertain (McCabe et al., 2007; Rössler et al., 2014).

Snowpack properties such as depth, density and snow water equivalent (SWE) can vary spatially and temporally across
the catchment landscape. Additionally, wind drift, landscape topography (i.e., slope, elevation, aspect) and vegetation cover (i.e., forest, grassland) affect the snowpack properties (Marks et al., 1998; Molotch et al., 2011; Stähli et al., 2000). Higher elevations are generally associated with greater snowpack depths due to higher precipitation rates and lower air temperatures (Beniston et al., 2003; Stewart, 2009). Compared to open grassland, forested landscapes tend to have shallower snow depths due to canopy interception of snowfall (Berris and Harr, 1987; López-Moreno and Stähli, 2007; Stähli and Gustafsson,
2006). Thus, snowpack outflow (snowmelt or a mixture of rainwater and snowmelt) is not generated homogeneously at the catchment scale (Berris and Harr, 1987; Würzer et al., 2016). Further, water flow paths within the snowpack can be highly variable, so that calculating or measuring the snowpack outflow can be challenging (Eiriksson et al., 2013; Kattelmann, 2000; Rücker et al., 2019; Unnikrishna et al., 2002; Webb et al., 2018).

A detailed understanding of snowpack outflow generation is needed at both the plot scale and the catchment scale in
order to quantify the runoff-contributing areas in snow-dominated catchments, so that runoff predictions during ROS events



are more reliable (DeWalle and Rango, 2008; Marks et al., 1998; Šanda et al., 2014). To track the heterogeneous contribution of snowpack outflow to streamflow during ROS events, environmental tracers can be used. Stable water isotopes ($\delta^{18}$O and $\delta^{2}$H) may be particularly useful as they allow streamflow to be separated into isotopically distinct sources (Klaus and McDonnell, 2013). Thus, if snowpack outflow is isotopically distinguishable from catchment groundwater

storage, its relative contributions to streamflow during ROS events can be quantified through two–component isotope-based hydrograph separation (IHS).

In some studies, the isotopic composition of bulk snow or of individual snow layers has been used as a proxy for snowmelt isotopic composition (Cooper et al., 1993; Dinçer et al., 1970; Huth et al., 2004; Maulé et al., 1994; Sueker et al., 2000). The isotopic composition of bulk snow is known to be variable in time and space, depending on catchment

characteristics such as latitude, exposure and elevation gradients (Dietermann and Weiler, 2013), as well as the structure of the forest canopy. Snowfall intercepted by the forest canopy is subject to sublimation, which is the main cause for the isotopic enrichment of winter throughfall (Claassen and Downey, 1995; Koeniger et al., 2008; Stichler, 1987). Although the spatial variations in the isotopic compositions of bulk snow and snowmelt are likely to be similar (Dietermann and Weiler, 2013), estimated meltwater contributions to streamflow can be significantly different when using bulk snow instead of

snowmelt as an endmember in IHS (Moore, 1989). Numerous studies have found that IHS in snow-dominated catchments is less uncertain when snowmelt is sampled directly (Obradovic and Sklash, 1986; Penna et al., 2017) with melt pans or snowmelt lysimeters (Beaulieu et al., 2012; Buttle, 1994; Hooper and Shoemaker, 1986b; Laudon et al., 2002; Schmieder et al., 2016; Shanley et al., 1995b; Unnikrishna et al., 2002; Wels et al., 1991). Snowmelt lysimeters also facilitate sampling at regular temporal intervals (i.e. daily, sub-daily), which is recommended because the isotopic composition of snowmelt can

be highly variable over time. This variability is caused by isotopic fractionation in the snowpack during phase changes (i.e., freezing and melting) (Judy et al., 1970; Lee et al., 2010b; Schmieder et al., 2016; Taylor et al., 2002b, 2001; Unnikrishna et al., 2002) and by ROS events when isotopically distinct rainwater percolates and mixes with the snowpack (Berman et al., 2009; Herrmann et al., 1981; Juras et al., 2016; Shanley et al., 1995a). Furthermore, isotopic exchange and redistribution in the snowpack can cause the snowpack outflow to be isotopically different from incoming rainfall (Judy et al., 1970; Lee et

al., 2010; Lee et al., 2010b; Taylor et al., 2001).

To the best of our knowledge, only two studies have estimated the contribution of snowpack outflow to streamflow during ROS events through using IHS (Buttle et al., 1995; Maclean et al., 1995). These studies used, however, only one or two ROS events that occurred before and during snowmelt. So far, the effects of both, vegetation and elevation on the generation of snowpack outflow and snowmelt's contribution to streamflow have not been investigated during ROS events.

To address these research gaps, we monitored snowpack outflow and its isotopic composition at three snowmelt lysimeter sites in the Alptal catchment in Central Switzerland. We measured snowpack outflow volumes at 10-min intervals during winter 2017 and 2018, as well as $\delta^{18}$O and $\delta^{2}$H in snowpack outflow at daily resolution during winter 2017. Because



the snowmelt lysimeter sites are located within an altitude range that includes the rain-transition zone and the monitored landscapes include forest and grassland vegetation, our setup allows for quantifying the spatio-temporal variability of snowpack outflow volumes and its effect on streamflow generation during ROS events.

## 2     Methodology

### 2.1     Field site

Field work was conducted in the southern part of the 47 km$^2$ Alptal catchment, 40 km south of Zürich in the northern pre-Alps in Central Switzerland (Figure 1). The Erlenbach catchment is a 0.7 km$^2$ tributary with a gauging station and a meteorological station.

In the Erlenbach catchment, winter precipitation is dominated by snowfall, which accounts for up to one-third of the total annual precipitation of roughly 2300 mm y$^{-1}$ (Feyen et al., 1999; van Meerveld et al., 2018). The annual average air temperature is 6 °C with a distinct seasonal cycle (-2 °C in February and 17 °C in August; Feyen et al., 1999). The Erlenbach catchment covers an altitude range from 1080 to 1520 m asl and frequent shifts between rain- and snowfall-dominated precipitation events occur during the winter season (Stähli and Gustafsson, 2006). The bedrock of the Erlenbach catchment is dominated by tertiary flysch, overlain by shallow soils with low permeability (Burch et al., 1996; Fischer et al., 2015). The landscape is characterized by coniferous forests (53 %), grassland (25 %) and a mixture of both (22 %) (Burch et al., 1996; Fischer et al., 2015; Keller, 1990).





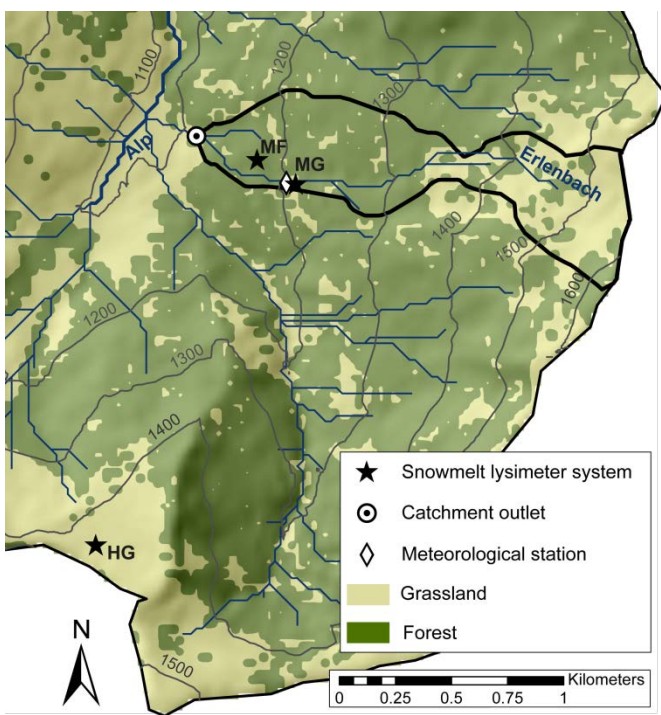

Source: Federal Statistical Office (BFS), GEOSTAT, CH-2010 Neuchâtel, Switzerland

**Figure 1: The Erlenbach catchment (heavy black outline) in the southern part of the Alptal valley, showing the distribution of vegetation (grassland and forest), as well as the locations of the three snowmelt lysimeter systems (MG: mid-elevation grassland site; MF: mid-elevation forest site; HG: high-elevation grassland site). A meteorological station is located near the MG site. At the Erlenbach catchment outlet, river discharge, precipitation (snow- and rainfall) and stable water isotopes in stream water and precipitation (snow- and rainfall) are measured.**

Two snowmelt lysimeter system sites, hereafter called MG (mid-elevation grassland) and MF (mid-elevation forest), are located at mid-elevations in the Erlenbach catchment at altitudes of 1216 and 1185 m asl, respectively. A third snowmelt lysimeter system was installed at a higher elevation (1405 m asl) site (the HG, or high-elevation grassland site), outside of the Erlenbach catchment near the southern border of the Alptal catchment, because an installation at this altitude within the Erlenbach catchment was technically not feasible. The MF site is dominated by a forest consisting of Picea abies and Abies alba, whereas the MG and HG sites are located on grassland. The MG and the MF sites are located 250 m apart from each other with an elevation difference of only 30 m. The terrain at the MG site is relatively flat, whereas the MF and HG sites are located on slightly sloping ridges.

At the meteorological station, we measured air temperature (107 Thermistor Probe, Campbell Scientific, Loughborough, Great Britain) every minute, as well as snow depth (Ultrasonic depth sensor, Judd Communications, Salt Lake City, Utah, USA) and precipitation (Lamprecht meteo GmbH, Rain gauge 15189, Göttingen, Germany) at 10-minute temporal resolution (Stähli and Gustafsson, 2006). Additionally, air temperature was measured at the HG and MF sites every 1 minute (107





Thermistor Probe, Campbell Scientific, Loughborough, Great Britain). At the Erlenbach catchment outlet, river discharge was recorded every 10-minutes (Burch et al., 1996) and a heated rain gauge (HOBO Rain Gauge, Metric Data Logger, RG3-M, Bourne, MA 02532, USA) measured precipitation at 10-minute intervals. The rain gauge at the meteorological station was not functioning during the period 18 February – 11 March 2017, and thus we filled the data gap with measurements

from another rain gauge, near the catchment outlet.

In the following analysis, we use the MG site as a reference site because it is located in close proximity to the meteorological station and it allows us to compare the effects of vegetation cover (MF vs. MG) and elevation (HG vs. MG).

## 2.2    A snowmelt lysimeter system for monitoring snowpack outflow and collecting water samples for stable water isotope analysis

The snowmelt lysimeter system was designed to measure the natural snowpack outflow and to collect samples for the analysis of stable water isotopes (see details in Rücker et al., 2019). The MG lysimeter system site was installed in March 2016, whereas the HG and MF lysimeter systems were installed in October and December 2016, respectively. Thus, the design of the HG and MF lysimeter systems could be slightly improved compared to that at the MG site.

Each of the three snowmelt lysimeter systems (MG, MF and HG) consists of three individual funnels (dimensions: 0.42
m diameter, 0.14 $m^2$ area, 0.059 m rim height) that are installed into the soil so that they collect the daily snowpack outflow at the snowpack-soil interface. From each individual lysimeter funnel, the snowpack outflow ran through a silicon rubber tube to a 10 L collection vessel. Every day at 05:40, an automatic water sampler (Maxx P6L – Vacuum System, Maxx GmbH, Rangendingen, Germany) pumped up to 300 ml of sample from the collection vessel into a dry 1-L HDPE autosampler bottle. After that, pinch valves at the lower outlet of the collection vessel were opened for 10 minutes to drain
the remaining liquid. When the snowpack outflow volumes reached the 10 L storage capacity of the water vessel, an additional bulk sample of each water vessel was collected by manually operating a pump cycle via a wireless connection. A whole pumping and rinsing cycle took around 20 minutes so that the starting time for the collection of the next water sample was set to 06:00. The filled sampling bottles were replaced with dry bottles once a week.

Before the snowpack outflow reached the water vessel, its volume was measured through a tipping bucket mechanism.
The tipping bucket mechanism was installed either directly below each individual lysimeter funnel (MG) or between the end of the silicon rubber tube and the collection vessel (MF, HG). The arrangement of the tipping bucket was changed for the MF and HG systems, so that it could easily be replaced or repaired if necessary. Because the tipping bucket mechanisms of each snowmelt lysimeter system were slightly adapted to the local properties at each field site, the average measurement uncertainties of the snowmelt outflow volume measurements were determined from replicate measurements of known
volumes poured into each funnel. The average measurement uncertainties differed by location and were 15 %, 7.5 % and 10 % at the HG, MG and MF sites, respectively.





In an earlier study that evaluated the performance of the snowmelt lysimeter design at the MG site, we found that the three individual funnels registered highly variable snowpack outflow volumes, thus reflecting temporal and spatial variability of the snowmelt processes at the plot scale (Rücker et al., 2019). In the analysis presented here, we averaged the snowpack outflow volumes collected with the three individual funnels at each lysimeter site. To express the uncertainty of

these measurements, we calculated the combined standard errors of these mean snowpack outflow volumes considering both the relative measurement uncertainty of the tipping bucket and the spatial variability of snowpack outflow generation. Measurements at 10-minute resolution were aggregated to daily resolution for the time period 06:00 till 05:40 of the following day. This time interval was chosen to correspond with the aggregation time used by the Federal Office of Meteorology and Climatology (MeteoSwiss). Ten-minute data were aggregated to hourly data over the periods HH:40 till

(HH+1):30 with HH denoting the hour of the measurement.

To prevent freezing in the tubes or in the tipping bucket mechanisms, a heating cable (Pentair, Raychem, BZV self-regulating heating band, Wisag, Fällanden, Switzerland) was attached to the silicon tubes of the MG and HG lysimeter systems in December 2017. In addition, a 12 W heating patch (110 mm x 77 mm) was attached next to the tipping bucket mechanism below the lysimeter funnel of the MG lysimeter system. The MF site was not equipped with either a heating

patch or a heating cable, because freezing was less problematic in the forest than at the open grassland sites. During winter 2018, the lysimeter funnels at the MG and HG sites were damaged due to the heavy snow cover. Thus, snowpack outflow volumes for the 2018 winter period were only available at the MF site.

At all three lysimeter sites, snow depths were measured with stakes located next to the individual lysimeter funnels. A webcam at each site recorded a picture of the stakes every hour, so that the snow depths could be determined from the

images. We used one image taken between 09:53 and 11:23 to estimate the daily snow depth at each lysimeter location. At the meteorological station near the MG site, the snow depth sensor provided additional measurements of maximum hourly snow depth, which were used to validate the daily data obtained from the webcam images. The snow depth sensor was not functioning during the period 8 March 2017 16:30 and 9 March 2017 00:50, and thus we filled the data gap with the snow depth estimates from the webcam pictures.

At the MG and HG sites, snow surveys were carried out at weekly intervals. In addition, snow surveys at monthly intervals were carried out at the MG and MF sites, so that the MG site was surveyed twice in the same week roughly once a month. During each survey, snow depth and bulk snow density were measured along a ca. 30-meter snow course with a snow tube (diameter 50 mm, length 1.2 m) to determine the snow water equivalent (SWE) (Stähli et al., 2000; Stähli and Gustafsson, 2006). Additionally, soil conditions were characterized (frozen or not frozen) with a 1-cm diameter aluminium

stake. At the MG site, the SWE prior to a ROS event was estimated by the product of the actual snow depth recorded by the snow depth sensor and the bulk density derived from the most recent snow survey. At the MF site, only two surveys were carried out in the winter of 2017.





### 2.2.1 Identification of rain-on-snow (ROS) events

ROS events during winter 2017 were identified based on the rainfall and snow properties at the meteorological station and the MG site. For winter 2018, measurements from the meteorological station and the MF site were used. The criteria for ROS events were: rainfall rates greater than 0.1 mm per hour, a total rainfall volume of at least 20 mm within 12 hours, air

temperatures above 0 °C and an initial snowpack depth of at least 10 cm.

To compare the volumes of snowpack outflow with those of incoming rainfall during the ROS events, we aggregated snowpack outflow volumes over the event periods. However, because the responses of snowpack outflow to a ROS event are generally delayed, the aggregation period for snowpack outflow was extended until the snowpack outflow volumes reached the pre-event flow rates or until a new rainfall event started. We defined the "snowpack water budget" of a ROS

event as the difference between snowpack outflow volume and rainfall volume.

### 2.2.2 Sample collection and isotope analysis

We measured stable water isotopes ($\delta^{18}O$, $\delta^2H$) in stream water, precipitation (snow- and rainfall), snowpack outflow and melted bulk snow samples. All of these samples were collected once each day except for the bulk snow, which was collected roughly once each week. At each of the three snowmelt lysimeter sites, daily composite samples of snowpack

outflow from each of the three individual funnels was collected with autosamplers for subsequent isotope analysis (Sect. 2.2). As shown in Rücker et al. (2019), the isotopic composition of snowpack outflow from the three individual funnels at each site were very similar, and thus we averaged the isotope values of snowpack outflow of the three individual lysimeter funnels at each site and expressed their spatial variability through the standard error of the mean.

Composite stream water samples were collected with an automatic water sampler (6712-Fullsize Portable Sampler,

Teledyne Isco, Lincoln (NE), USA) that pumped 100 ml of stream water into a dry 1-L HDPE bottle four times a day (at 05:40, 11:40, 17:40, and 23:40). Every three weeks, the filled autosampler bottles were replaced with empty ones. Stable water isotopes in precipitation were measured at hourly resolution directly at the Erlenbach catchment outlet with a wavelength-scanned cavity ring-down spectrometer (CRDS; model L2130-I; Picarro Inc., Santa Clara, CA, USA) coupled to Picarro Inc.'s Continuous Water Sampler module. The analytical uncertainty of the analyser was 0.09 ‰ $\delta^{18}O$ and 0.21 ‰

for $\delta^2H$ (von Freyberg et al., 2018). More details about the high-frequency sampling approach can be found in von Freyberg et al. (2017) and von Freyberg et al. (2018). To obtain daily precipitation isotope values, these hourly precipitation isotope measurements were volume-weighted with hourly precipitation volumes over the time period 06:00 till 05:00 of the following day. Whenever possible, this volume-weighting was based on precipitation measurements from the meteorological station at the MG site; however, during the period of instrument failure (18 February – 11 March 2017),



precipitation measurements from the heated rain gauge at the catchment outlet were used instead (i.e., for ROS events #4 and #5).

During the snow surveys, bulk snow was collected from the entire snow profile close to the lysimeter sites with a snow tube (diameter 50 mm, length 1.2 m) and transferred to a HDPE plastic bag (300 x 500 x 0.1 mm, Plasti-Pac Zürich AG, Zürich, Switzerland), which was sealed immediately. At the MG site, one or two bulk snow samples were collected near the lysimeter funnels at two different days every week during the snow surveys in winter 2017. Because there was generally more snow at the HG site, three bulk snow samples were collected there on the same day once per week. The isotopic compositions of these three bulk snow samples were averaged to obtain one weekly mean value for the HG site.

All water samples that were collected in the field were stored in sealed bottles and refrigerated at 4 °C in the laboratory until sample preparation. Frozen samples were melted in the laboratory at room temperature. All samples were filtered through 0.45-µm Teflon filters (DigiFilter micron Teflon, S-Prep GmbH, Überlingen, Germany) and filled into 2-ml glass vials with silicon seals. Isotope concentrations were measured with an LGR IWA-45EP off-axis integrated cavity output spectrometer (ABB Los Gatos Research, San Jose, California, USA) at the laboratory of the Swiss Federal Institute for Forest, Snow and Landscape Research (WSL). Isotopic abundances are reported using the δ notation relative to the IAEA-VSMOW-II and SLAP-II standards. The analytical uncertainty of the analyser was 0.21 ‰ for $\delta^{18}$O and 0.37 ‰ for $\delta^2$H, which was estimated from replicate check-standard measurements within the same batch.

### 2.3 Two-component isotope-based hydrograph separation

We used two-component isotope-based hydrograph separation (IHS) to estimate the relative contribution of snowpack outflow to catchment streamflow. The calculation of the relative contribution of snowpack outflow to streamflow ($F_{spo}$) was based on the conventional mass balance equation of Pinder and Jones (1969):

$$F_{spo} = \frac{V_{spo}}{V_S} = \frac{c_S - c_{pe}}{c_{spo}^* - c_{pe}},$$ (1)

where $V_{spo}$ denotes the volume of snowpack outflow in streamflow ($V_S$), and $C_S$, $C_{pe}$ and $C_{spo}^*$ denote the tracer concentrations in stream water, pre-event water and snowpack outflow, respectively. We used daily time steps for all calculations. The pre-event tracer concentration ($C_{pe}$) was represented by the isotopic composition of stream water of the day prior to the ROS event of interest and was assumed to be constant during the event. The isotopic composition of snowpack outflow at day $i$ was calculated as the incremental volume-weighted mean using the measured volumes of snowpack outflow or rainfall since the beginning (index $k=j$) of the event (McDonnell et al., 1990):

$$C_{spo,i}^* = \frac{\sum_{k=j}^{i}(V_{spo,k}C_{spo,k})}{\sum_{k=j}^{i}V_{spo,k}} .$$ (2)





Equation (3) quantifies the standard error of $C_{spo}^*$ ($SE_{C_{spo}^*}$), which is a combination of the measurement uncertainty of the isotope analyser (first summand on the right side) and the spatial variability of the snowpack outflow volumes collected by the three individual funnels of each lysimeter system (second summand on the right side):

$$SE_{C_{spo,i}^*} = \sqrt{\sum_{k=j}^{i}\left(\frac{V_{spo,k}}{\sum_{k=j}^{i}V_{spo,k}}SE_{C_{spo,k}}\right)^2 + \sum_{k=j}^{i}\left[\left(\frac{C_{spo,k}}{\sum_{k=j}^{i}V_{spo,k}} - \frac{\sum_{k=j}^{i}(C_{spo,k}V_{spo,k})}{\left(\sum_{k=j}^{i}V_{spo,k}\right)^2}\right)SE_{V_{spo,k}}\right]^2} \; . \tag{3}$$

In Eq. (3), $SE_{C_{spo}}$ is the standard error of the isotope data, which is assumed to be the measurement uncertainty of the isotope analyser, and $SE_{V_{spo}}$ is the standard error of the mean of the three individual snowpack outflow volumes measured with the individual lysimeter funnels. Using Gaussian error propagation, the uncertainty of $F_{spo}$ was estimated as

$$SE_{F_{spo}} = \sqrt{\left(\frac{-1}{(C_{pe}-C_{spo}^*)}SE_{C_S}\right)^2 + \left(\frac{C_S - C_{spo}^*}{(C_{pe}-C_{spo}^*)^2}SE_{C_{pe}}\right)^2 + \left(\frac{C_{pe}-C_S}{(C_{pe}-C_{spo}^*)^2}SE_{C_{spo}^*}\right)^2} \; . \tag{4}$$

We assume that the standard errors of $C_S$ and $C_{pe}$ ($SE_{C_S}$ and $SE_{C_{pe}}$) are equivalent to the measurement uncertainty of the
isotope analyser.

Since we measured snowpack outflow volumes and their isotopic compositions only at three locations and not across the entire catchment, we cannot reliably estimate the catchment-wide snowpack outflow contribution during individual ROS events. Instead, we performed IHS for each ROS event and individually for each sampling site by using the site-specific measurements of snowpack outflow volume and isotopic composition. Thus, we obtained the relative contributions of
snowpack outflow to streamflow for three different scenarios during winter 2017, under the assumption that the catchment-wide average snowpack outflow is represented either by the measurements from the mid-elevation grassland (MG), the mid-elevation forest (MF) or the high-elevation grassland (HG) site. By comparing the IHS results of the three scenarios for each ROS event, we seek to quantify the effects of spatial variability in snowpack outflow generation due to vegetation and elevation. Since no snowpack outflow could be measured at the HG and MG sites during winter 2018, a three-scenario
comparison was not possible for that period.

We also quantified the relative contributions of rainwater (subscript $R$) and pre-event water to streamflow with

$$F_R = \frac{V_R}{V_S} = \frac{C_S - C_{pe}}{C_R^* - C_{pe}} \quad , \tag{5}$$

Where $V_R$ and $C_R^*$ denote the rainfall volume and the volume-weighted isotope concentration in rainwater, respectively. The standard error of $C_R^*$ ($SE_{C_R^*}$) was estimated with:

$$SE_{C_{R,i}^*} = \sqrt{\frac{\sum_{k=j}^{i}V_{R,k}\left(C_{R,k} - C_{R,k}^*\right)^2}{(j-i)\sum_{k=j}^{i}V_{R,k}}} \quad . \tag{6}$$



To quantify the standard error of $F_R$ ($SE_{F_{spo}}$) we used Eq. (4), in which we replaced $SE_{C^*_{spo}}$ with $SE_{C^*_R}$ and $C^*_{spo}$ with $C^*_R$. We determined $F_R$ based on the measurements collected at the meteorological station of the Erlenbach catchment. Because snowpack outflow volumes and isotopic compositions were also measured at the same location (i.e., MG site), we can compare $F_{spo}$ with $F_R$ to study the role of snowpack storage for streamflow generation. We expect that mixing processes and storage of incoming rainfall in the snowpack result in a more damped isotope signal of snowpack outflow compared to the isotope signal of incoming rainfall. In this case and all else equal, the fraction $F_{spo}$ will be larger than the fraction $F_R$.

## 3      Results and Discussion

### 3.1      Variable snow conditions at the three snowmelt lysimeter sites and response of discharge

#### 3.1.1      Spatial and temporal variability of snow properties due to elevation and vegetation

During winter 2017 (1 January – 7 May 2017), most of our study catchment was covered with a seasonal snowpack. In the beginning of January 2017, when snowfall occurred over several consecutive days during cold conditions, the seasonal snowpack established simultaneously at all three snowmelt lysimeter sites (HG, MG and MF; Figure 2a-c). Figure 2 shows that the snow depths at the three sampling sites differed from one another and varied considerably over time. Thus, in the following analysis, we used the mid-elevation grassland (MG) site as a reference location to compare the snowpack outflow volumes of the mid-elevation forest (MF) site and high-elevation grassland (HG) site.

At the MG site, snow depth was highest on 17 January 2017 (82.2 cm) and SWE was greatest on 20 February 2017 (168 mm; Figure 2b). The seasonal snow cover was established on 3 January, became discontinuous on 17 March 2017 and was melted completely by 20 March 2017. Two short-term snowpacks established during additional snowfall events in mid and late April 2017. The snow depth measurements at 10-minute and daily resolution agree well for most of the study period except for the last 3 weeks of the seasonal snowpack (1-24 March 2017). For this period, the daily snow depth readings from three measurement stakes indicated lower mean snow depths compared to the readings of the snow depth sensor (10-minute data). These measurement differences can be explained by small-scale spatial heterogeneities of the seasonal snowpack caused by wind drift or enhanced melt around the measurement stakes.

Compared to the MG site, maximum snow depth at the 200-meter higher HG site was reached about seven weeks later and was 55 cm greater (137 cm; Figure 2a). The maximum SWE (303.4 mm) occurred about nine days after the peak in snow depth, and was almost two-times larger than the maximum SWE at the MG site. According to Stähli et al. (2000) and Stähli and Gustafsson (2006) snow depths, and thus SWE, were generally larger at higher elevations in the Alp catchment due to lower temperatures and thus a greater tendency for winter precipitation to fall as snow. Due to the greater snow depth



at the HG site, the seasonal snowpack lasted around 21 days longer than at the MG site. Similar to the seasonal snowpack, the two short-term snowpacks in mid and late April 2017 reached greater snow depths and SWEs compared to the MG site.

At the forested MF site, the snowpack was generally much thinner compared to the nearby grassland MG site, and thus melted out several times during our study period (Figure 2c). The maximum snowpack depth at the MF site was around

30 cm lower than at the nearby MG site. Based on monthly surveys, the largest SWE (72 mm) occurred on 25 January 2017, roughly one month earlier than the maximum at the MG site. Because of the generally smaller snow depths at the MF site, its seasonal snowpack became discontinuous on 21 February 2017, which was 24 days earlier compared to the MG site. Several snowfall events (24 February and 6 March 2017, 18 April and 27 April 2017) resulted in shallow snowpacks at the MF site that lasted only several days (Figure 2c). An earlier study in the Alp catchment observed that roughly twice as much

snow accumulated at grassland sites than at nearby forested sites (Stähli et al., 2000). Snow accumulation under forest is often significantly smaller due to interception and canopy effects on radiation (i.e. lower shortwave and higher longwave radiation; Berris and Harr, 1987; Bründl, 1997; Gustafson et al., 2010; López-Moreno and Stähli, 2007; Molotch et al., 2011; Montesi et al., 2004).

Total volumes of snowpack outflow, cumulated over the entire study period, were largest at the HG site (1319±214 mm)

and smallest at the MF site (685±78 mm). At the MG site, cumulative volumes of snowpack outflow (816±128 mm) were similar to cumulative volumes of incoming rainfall (833±17 mm) and discharge at the catchment outlet (786 mm).

Weekly snow surveys at the MG site showed that the shallow soil was frozen between 28 December 2016 and 12 March 2017, likely because air temperatures were mostly below 0 °C before the seasonal snowpack was established and the seasonal snow cover prevented the soil frost from thawing despite warmer conditions until mid-March (Goodrich, 1982).

At the MF site, soil frost was monitored monthly and the only survey that indicated soil frost was on 28 December 2016, i.e. before the seasonal snowpack was established. On 25 January 2017, the shallow soil at the MF site was no longer frozen.



**Figure 2:**
**Measurements of daily precipitation (snow-and rainfall) volumes measured at the MG site during winter 2017, as well as hourly air temperature, snow depth, snow water equivalent (SWE) and snowpack outflow volumes measured at the (a) high-elevation grassland site (HG, red), (b) mid-elevation grassland site (MG, yellow), and (c) mid-elevation forest site (MF, green) for the study period 1 January – 22 May 2017. Panel (d) shows daily discharge at the Erlenbach catchment outlet (on log scale). Vertical grey**





bars indicate the six rain-on-snow (ROS) events that are analysed in this study. The asterisk (*) indicates data gaps. Data of winter 2018 are shown in the supplement.

### 3.1.2    The spatially variable response of snowpack outflow to rain-on-snow (ROS) events

The study period was characterized by frequent ROS events, which altered the snowpack properties at the three snowmelt lysimeter sites during the winter of 2017. Six ROS events are discussed in detail below to compare the snowmelt processes at the HG, MG and MF sites. A further four ROS events occurred during winter 2018, but only the MF site provided data of snowpack outflow volumes, so that a site-to-site comparison was not possible.

Figure 2 and Table 1 provide overviews of the six ROS events during winter 2017 (events #1-#6) and the four events during winter 2018 (only MF: events #7-#10). During ROS event #1, the tipping bucket rain gauge at the meteorological station stopped working after 13 January 2017 03:40, which coincided with an air temperature decrease to values below 0 °C (Figure 2). At that point, 21.6 mm of rainfall had been recorded since the beginning of the ROS event on 12 January 2017 17:40. Our webcam images and snow depth data show, however, that snow depth increased after 03:40, indicating the transition from rainfall to snowfall. Also, river discharge peaked soon afterwards at 05:40, providing further evidence of a catchment-wide transition from rain to snow. Thus, despite the malfunctioning of the tipping bucket rain gauge, we consider the rainfall measurements until 03:40 to be representative for the total volume of incoming rainfall during ROS event #1.

**Table 1: Start and end times of the ten rain-on-snow (ROS) events that were identified in the winters of 2017 and 2018 (only MF site). No reliable snowpack outflow volumes were measured at the HG and MG site in 2018. Rainfall characteristics (measured at the meteorological station near the MG site), such as cumulative rainfall of the ROS event, maximum 4-hour rainfall volume, maximum 8-hour rainfall volume and rainfall duration and snowpack outflow volumes at the three snowmelt lysimeter sites HG, MG and MF are shown. The standard error (SE) of the snowpack outflow measurements represents the combined effect of measurement uncertainty and spatial heterogeneity of the melt process at each sampling site.**

| ROS event number | Start time (d.m.y hr:min) | End time (d.m.y hr:min) | Rain-fall (mm) | Maximum 4-h rainfall (mm) | Maximum 8-h rainfall (mm) | Rainfall duration (h) | HG site | MG site | MF site |
|---|---|---|---|---|---|---|---|---|---|
| | | | | MG site | | | Snowpack outflow volumes ± SE (mm) | | |
| #1 | 12.1.2017 17:40 | 16.1.2017 14:40 | 21.6 | 10.2 | 20.1 | 10 | [a] | 2.5±1.3 | 10.1±2.4 |
| #2 | 30.1.2017 15:40 | 2.2.2017 04:40 | 99.2 | 19.4 | 29.6 | 44 | 125.7±21.6 | 67.8±44.5 | 87.1±16.8 |
| #3 | 21.2.2017 04:40 | 22.2.2017 10:40 | 20.0 | 5.6 | 5.8 | 22 | 62.3±34.7 | 22.5±7.2 | 46.5±10.3 |
| #4 | 1.3.2017 18:40 | 3.3.2017 01:40 | 33.6 | 10.4 | 19.8 | 19 | [a] | 29.6±5.1 | 35.0±4.2 |
| 5 | 8.3.2017 16:40 | 10.3.2017 11:40 | 90.2 | 19.0 | 34.8 | 5 | 16.2±13.5 | 22.0±13.1 | 133.2±18.4 |
| #6 | 18.3.2017 06:40 | 19.3.2017 20:40 | 66.9 | 29.1 | 36.0 | 27 | 58.2±19.8 | 108.9±26.1 | 41.1±5.6 |
| #7 | 3.1.2018 22:20 | 5.1.2018 19:50 | 53.0 | 12.1 | 20.8 | 33 | - | - | 61.0±42.7 |
| #8 | 20.1.2018 17:50 | 21.1.2018 14:00 | 34.6 | 15.9 | 26.5 | 14 | - | - | 55.0±13.0 |





| | | | | | | | | | |
|---|---|---|---|---|---|---|---|---|---|
| #9 | 21.1.2018 23:10 | 23.1.2018 21:10 | 129.4 | 23.0 | 45.5 | 33 | - | - | 159.3±32.6 |
| #10 | 15.2.2018 11:10 | 17.2.2018 05:40 | 59.8 | 19.9 | 24.8 | 30 | - | - | 54.1±11.5 |

[a] no snowpack outflow occurred

Figure 3a compares the cumulative volumes of rainfall and snowpack outflow (MG, MF and HG site) of the ten ROS events during winter 2017 and 2018 (only MF site). The snowpack outflow volumes measured at the three snowmelt lysimeter sites were often associated with large uncertainties (Figure 3a, b), mainly because snowmelt at the plot scale can be

very heterogeneous (Kattelmann, 2000; Rücker et al., 2019; Unnikrishna et al., 2002). Because each sampling site consists of three individual lysimeters, we were able to estimate (at least approximately) this spatial variability of snowpack outflow.

At the MG site, the snowpack response to ROS events was highly variable (Figure 3b), i.e., snowpack outflow was less than incoming rainfall (events #1 and #5), similar to incoming rainfall (events #2, #3 and #4) or more than incoming rainfall (event #6). For events #1, #5 and #6 the differences between rainfall volumes and snowpack outflow were statistically

significant (i.e. larger than two times their pooled standard errors).

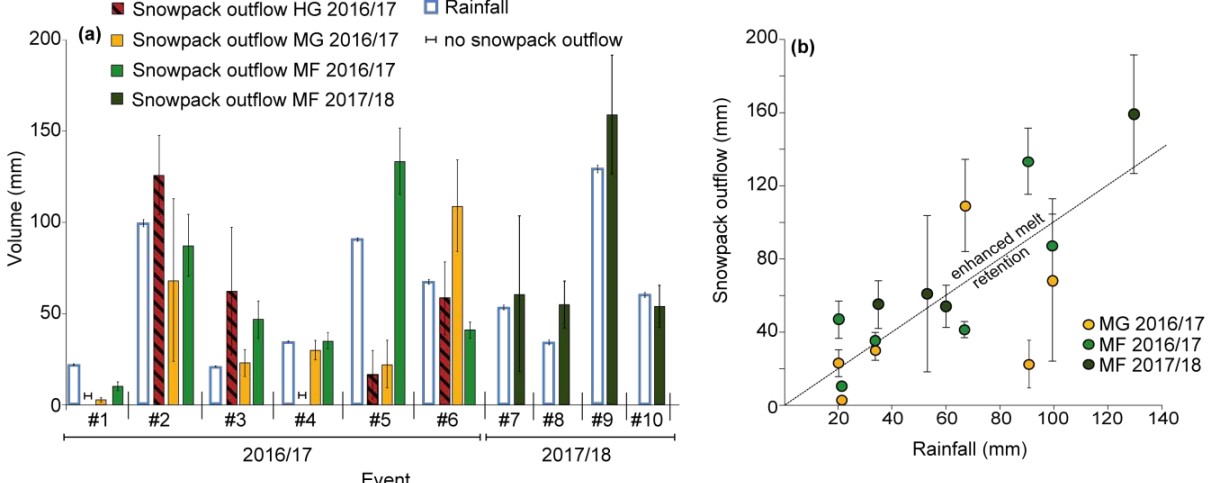

**Figure 3:** Comparison of the 10 rain-on-snow (ROS) events of winter 2017 (HG, MG and MF site) and 2018 (event #7-#10 only MF site) indicate large spatial and temporal variability of snowpack outflow generation in response to incoming rainfall. (a) Rainfall

volumes at the MG site (light blue) and snowpack outflow volumes at the HG (red, black-shaded), MG (yellow) and MF (light green: winter 2017; dark green: winter 2018) sites. Error bars indicate the standard error (SE) of the snowpack outflow measurements (combining measurement uncertainty and spatial heterogeneity of the melt process at each sampling site). (b) Comparison of snowpack outflow volumes and rainfall volumes during the 10 ROS events at the MG and MF sites (colour coding as in Fig. 3a). ROS events with enhanced melt plot above the 1:1 line, whereas events with rainfall retention in the snowpack plot

below the 1:1 line. Error bars indicate ±SE. Please note that the MF site was already snow-free during event #6.

At the 200-meter higher HG site, snowpack outflow volumes were similar to those measured at the MG site (within their pooled standard errors), except for event #4 and #6. During event #4, no snowpack outflow was generated at the HG site,





probably because the local air temperature was lower and the snowpack was 97 cm deeper and thus retained more rainwater than the snowpack at the MG site (Figure 2a). Similarly, less snowpack outflow was recorded at the HG site than at the MG site during ROS event #6, probably because the deeper HG snowpack was not yet saturated. For event #2, however, the measurement differences between the three individual lysimeters at the MG site were particularly large, likely due to lateral flow in the snowpack (Eiriksson et al., 2013; Webb et al., 2018).

The MF and MG sites are located close to each other, so that deviations in snowpack outflow can be attributed to effects of vegetation cover (grassland vs. forest). For the ROS events #1 and #5, snowpack outflow at the forested MF site was larger compared to the grassland MG site, whereas it was smaller for event #6. For the remaining events, the differences between incoming rainfall and snowpack outflow volumes were not statistically significant. Note that for events #3 and #4 the MF site had snowpacks of only 5 cm and 8 cm, respectively. Larger snowpack outflow volumes (events #1 and #5) at the MF site can be explained by the shallower snow depths below the forest canopy (Figure 2a). Hence, the shallower snowpack saturated more rapidly during ROS events and additional meltwater was released, so that more snowpack outflow was generated compared to the MG site where the snowpack was deeper (Berg et al., 1991; Berris and Harr, 1987; Wever et al., 2014). During event #6, the MF site was already snow-free so that the lysimeter funnels captured only throughfall, which was less than the rainfall volumes measured near the MG site due to interception losses (DeWalle and Rango, 2008; Saxena, 1986). For the ROS events #7 to #9, the snowpack outflow volumes of the MF site were larger compared to the incoming rainfall volumes, thus indicating enhanced melt. The highest snowpack outflow volumes during winter 2018 were registered during event #9 which followed one day after ROS event #8. The snowpack at the beginning of event #9 was thinner (event #8: 23 cm; event #9: 10 cm) and probably more saturated, with a higher snow bulk density compared to event #8.

The detailed analysis of the six ROS events at the three lysimeter sites during winter 2017 shows that incoming rainfall was attenuated differently in the snowpacks (both among sites and among events), illustrating the challenge of adequately estimating snowpack outflow volumes during ROS events at the plot and catchment scale. Previous studies used rainfall characteristics and snowpack properties to predict the effects of ROS events on catchment outflow (DeWalle and Rango, 2008; Kattelmann, 1997; Würzer et al., 2016). Thus, in the following section, we analyse the processes and properties that control the outflow response of the snowpack.

### 3.1.3 The effects of snowpack properties and rainfall characteristics on snowpack outflow generation during ROS events

Figure 4a-c compare the snowpack water budgets of the ROS events with initial snow properties (i.e., bulk snow density, SWE and snow depth) and rainfall characteristics (maximum cumulative 4-h rainfall, maximum cumulative 8-h rainfall, event duration and mean air temperature) to better understand their effects on snowpack outflow generation. The snowpack



water budget was calculated as the volumetric difference between snowpack outflow and incoming rainfall, so that positive values of the snowpack water budget indicate enhanced snowmelt whereas negative values indicate retention of incoming rainfall in the snowpack (Table 1). Note that the MF site was already snow-free prior to ROS event #6, and thus we excluded this data point from the following analysis. At the MG site, ROS event #6 had the most positive snowpack water budget (42.0±26.1 mm); i.e. snowpack outflow was 1.6-times larger than incoming rainfall. The most positive snowpack water budgets at the MF site occurred during events #5 (winter 2017) and #9 (winter 2018).

Figure 4 shows that most of the relationships between the snowpack water budgets and initial snow properties or rainfall characteristics are highly scattered, likely because different processes control the flow of rainwater through the snowpack at different times and places. Some of the events indicate that more snowpack outflow is generated when initial snowpack density is higher (Figure 4a) or when rainfall intensities are lower (Figure 4d). When additional data of winter 2018 from the MF site are considered, a linear relationship ($R^2$=0.51) in Figure 4c suggests that shallower initial snowpacks were associated with enhanced melt. No consistent relationships emerge between snowpack water budget and initial SWE (Figure 4c) or rainfall duration (Figure 4e), implying that these are poor predictors for snowpack responses to ROS events. This is further confirmed by a multiple linear regression analysis (Software JMP 14, 100 SAS Campus Drive, Cary, NC 27513, USA) that was used to quantify the effects of initial snowpack properties and rainfall conditions on snowpack outflow volumes for the 15 ROS events measured at the MG and MF sites. The best model fit (n=15, RMSE = 22.097; $R^2$=0.8; $R^2$adjusted=0.76) was obtained with two predictor variables, initial snowpack depth prior to the event ($p$-value < 0.0001) and rain volume ($p$-value=0.004). The effects of other variables (rainfall intensity, rainfall duration, mean air temperature during event) on snowpack outflow volumes were not significant. A better understanding of snowmelt processes during ROS events can be obtained when individual events or event pairs are analysed in greater detail.

Snowmelt at the MG site was most enhanced during event #6, when 66.9 mm of rainfall resulted in 63±39 % (42.0±26.1 mm) more snowpack outflow (total outflow 108.9±26.1 mm). This ROS event occurred during the melt-out phase of the seasonal snowpack at the MG site, when it was isothermal, ripe and already melting (high density 0.402 g cm$^{-3}$, small snow depth 16.9 cm, low SWE 67.9 mm; Table 2). At the end of the ROS event, the snowpack was entirely melted. In addition, rainfall intensities during event #6 were the highest of all six events (maximum 4-hour rainfall: 29.1 mm, maximum 8-hour rainfall: 36 mm). As a result, the time lag of snowpack outflow to incoming rainfall was short (Figure 5) and incoming rainfall accelerated the melt-process (Berg et al., 1991; Colbeck, 1977; MacDonald and Hoffman, 1995; Marks et al., 1998; Wever et al., 2014).

The 90.2 mm ROS event #5 resulted in the most negative snowpack water budget (-68.4±13.1 mm) at the MG site, with 76±14 % of incoming rainfall being retained in the snowpack. By comparing event #5 to event #2, during which a similar amount of rain fell (99.2 mm) and the average snowpack water budget was less negative (-31.4±44.5 mm or 32±45 %), we find that the mean air temperature during these events were similar (4.43 °C and 3.14 °C, respectively; Table 2). However,



the main difference between both events was the higher air temperature prior to event #2 (5.1 °C) compared to event #5
(1.6 °C). This suggests that the cold content of the snowpack prior to event #5 was higher due to initially colder atmospheric
conditions and that more rainfall froze and was retained in the snowpack compared to event #2 (Berghuijs et al., 2014;
DeWalle and Rango, 2008; Juras et al., 2016; Maclean et al., 1995; Marks et al., 1998; Wever et al., 2014).

**Table 2: Event characteristics and snowpack water budgets of the ten rain-on-snow (ROS) events. Snowpack water budgets were
calculated for the MG and the MF site by subtracting the snowpack outflow volume from the rainfall volume. Standard errors
(SE) of the snowpack water budget were estimated from the measurement uncertainty and the spatial variability of snowpack
outflow measurements at the sampling site. Manual snow surveys provided initial snow bulk density and initial snow water
equivalent (SWE). Snow surveys were generally performed once or twice a week at the MG site but only monthly at the MF site
during the winter of 2017, providing insufficient information at that site (-). Weekly snow surveys at the MF site in winter 2018
did not properly represent the snow properties at the MF site (-) because the snow depth under the forest canopy was much
shallower and highly variable over time compared to the MG site, and thus measured snow bulk densities at the MF site were
likely not constant over several consecutive days.**

| ROS event number | Field site | ROS event start time (d.m.y hr:min) | Date of snow survey (d.m.y) | Initial SWE from survey (mm) | Initial snow bulk density from survey (g cm$^{-3}$) | Initial snow depth from webcam or snow depth sensor (cm) | Mean air temperature (°C) during event | Snowpack water budget ±SE (mm) |
|---|---|---|---|---|---|---|---|---|
| #1 | MG | 12.1.2017 17:40 | 9.1.2017 | 39.78 | 0.152 | 26 | 1.1 | -19.1±1.3 |
| #2 | MG | 30.1.2017 15:40 | 30.1.2017 | 127.59 | 0.262 | 49 | 3.1 | -31.4±44.5 |
| #3 | MG | 21.2.2017 04:40 | 20.2.2017 | 135.14 | 0.343 | 39 | 5.0 | 2.4±7.2 |
| #4 | MG | 1.3.2017 18:40 | 27.2.2017 | 157.35 | 0.363 | 54 | 2.7 | -4±5.1 |
| #5 | MG | 8.3.2017 16:40 | 6.3.2017 | 138.22 | 0.312 | 44 | 4.4 | -68.4±13.1 |
| #6 | MG | 18.3.2017 06:40 | 12.3.2017 | 67.94 | 0.402 | 17 | 5.3 | 42±26.1 |
| #1 | MF | 12.1.2017 17:40 | - | - | - | 18 | 0.8 | -11.5±2.4 |
| #2 | MF | 30.1.2017 15:40 | - | - | - | 29 | 2.7 | -12.1±16.8 |
| #3 | MF | 21.2.2017 04:40 | - | - | - | 5 | 4.0 | 26.5±10.3 |
| #4 | MF | 1.3.2017 18:40 | - | - | - | 8 | 2.3 | 1.4±4.2 |
| #5 | MF | 8.3.2017 16:40 | - | - | - | 20 | 3.9 | 43±18.4 |
| #6 | MF | 18.3.2017 06:40 | - | - | - | 0 | 6.5 | -25.8±5.6 |
| #7 | MF | 3.1.2018 22:20 | - | - | - | 31 | 4.4 | 61±42.7 |
| #8 | MF | 20.1.2018 17:50 | - | - | - | 23 | 1.2 | 55±13 |
| #9 | MF | 21.1.2018 23:10 | - | - | - | 10 | 2.5 | 159.3±32.6 |
| #10 | MF | 15.2.2018 11:10 | - | - | - | 22 | 3.4 | 54.1±11.5 |

A similar analysis can be carried out for ROS events #1 and #3, during which 21.6 mm and 20.0 mm of rain fell,
respectively. During event #1, snowpack outflow volumes at the MG site were 88±6 % (19.1±1.4 mm) less than rainfall,
indicating significant retention of rainwater in the snowpack. The air temperature was low before this event (0.4 °C),
resulting in a less dense snowpack with a high cold content, which could retain more rainfall by freezing (DeWalle and
Rango, 2008). During event #3, however, the snowpack water budget was very small (12±36 % or 2.45±7.2 mm),





suggesting that either most of the rainwater percolated through the snowpack, that incoming rainwater was stored in the snowpack and replaced an equal volume of meltwater, or a combination of both. The initial conditions of event #3 differed to those of event #1, with rainfall duration being twice as long (22 h), initial air temperature being 3.7 °C higher and a nearly two-times denser snowpack (0.34 g cm$^{-3}$; Figure 4). Thus, although the initial snowpack of event #3 was slightly deeper

(39.4 cm) than that of event #1 (26.2 cm), the longer rainfall duration, higher air temperatures and denser snowpack reduced the storage capacity in the snowpack and less rainfall was retained. The significant role of rainfall duration in snowpack outflow generation during ROS events has previously been shown by Kattelmann (1997) and Kroczynski (2004).

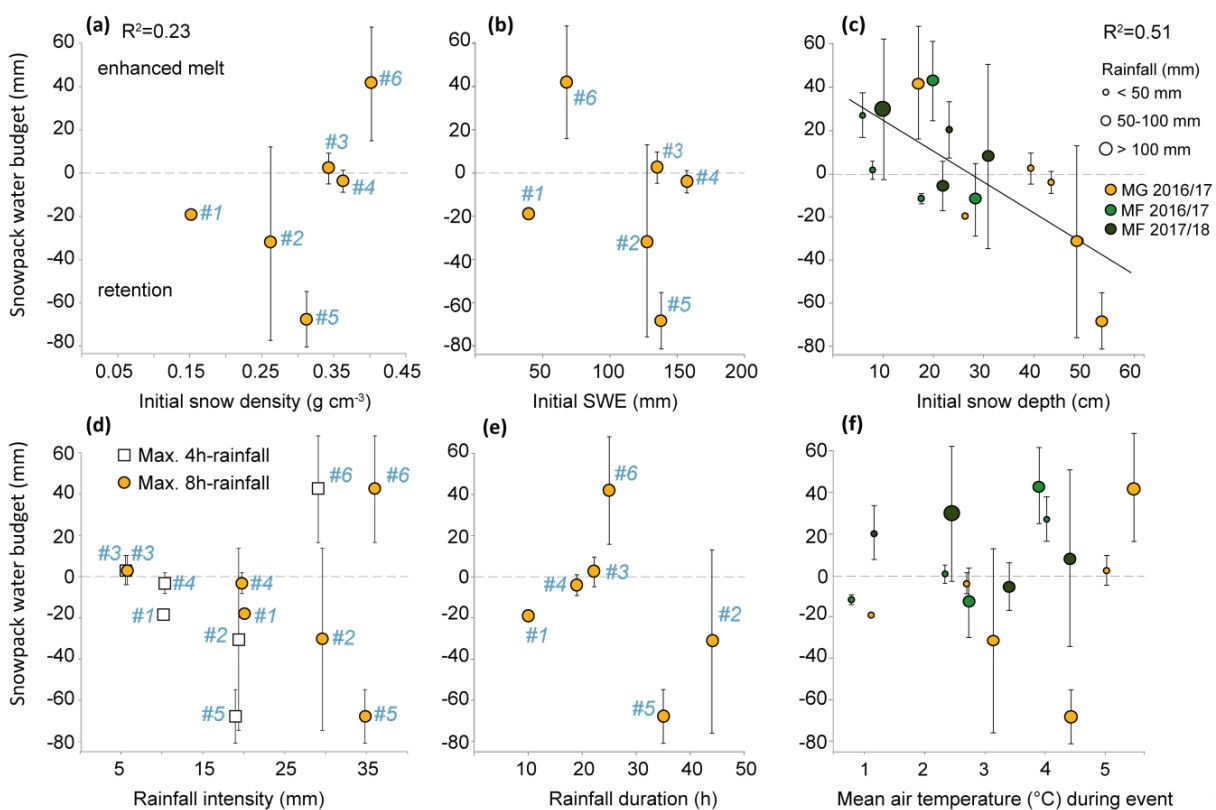

**Figure 4:**

**The correlation of snowpack water budgets (snowpack outflow subtracted by rainfall volume) with initial snow conditions and rainfall characteristics (measured at MG site) of the rain-on-snow (ROS) events show the strongest relationship with initial snow depth. Positive values of the snowpack water budget indicate enhanced snowmelt, whereas negative values indicate retention of incoming rainfall in the snowpack. Error bars indicate the uncertainty of the snowpack outflow, i.e. the combined effects of measurement uncertainty and spatial variability. The different scatter plots compare the snowpack water budgets at the MG site**

**of winter 2017 with (a) initial snow density, (b) initial snow water equivalent (SWE), (c) initial snow depth, (d) rainfall intensity presented as maximum 4-hour and 8-hour rainfall volumes, (e) rain duration, and (f) mean air temperature (°C) during the ROS event. In Figure (c) and (f) data from the MF site are included (light green: winter 2017; dark green: winter 2018), and the size of the scatter points indicate the rainfall volume (mm) on event basis.**



The analysis of the ten ROS events measured at the MG and MF site illustrates that the generation of snowpack outflow does not entirely depend on the incoming rainfall volume, but also on the initial snowpack conditions that control retention of rainfall and melt processes. This is further illustrated by the hourly measurements of snowpack outflow that indicate highly variable responses and lag times across the lysimeter sites (Figure 5). During four ROS events (#1, #2, #3 and #5),

snowpack outflow occurred much earlier at the MF site than at the MG and HG sites (Figure 1). Because the snowpacks at the MG and HG sites were generally deeper and less saturated than at the MF site, their snowmelt volumes were smaller and their snowpack outputs were delayed. Thus, the magnitude and timing of snowmelt at the catchment scale strongly depend on the depth and the degree of ripeness of the snowpack (Berg et al., 1991; DeWalle and Rango, 2008; MacDonald and Hoffman, 1995; Maclean et al., 1995; Wever et al., 2014).

Measurements from the MG, MF and HG sites reveal that snowpack outflow generation is highly variable across spatial and temporal scales and as a result, the contribution of snowpack outflow to river streamflow is very heterogeneous across the catchment landscape. For instance, streamflow response to ROS event #2 was particularly large, likely because of large snowmelt inputs from higher elevations (HG site; Figure 5b). Daily pulses of snowmelt from the HG site in late March were also reflected in distinct diurnal variations in stream discharge, suggesting input of snowmelt mainly from high elevations

(Figure 5b). In contrast, during events #1 and #4, no snowmelt was generated at the HG site, so that the observed discharge peak was likely to be caused by snowmelt from low and mid elevations (MG and MF sites; Figure 5c and d). However, the synchrony of responses does not allow drawing any conclusions about the water sources of streamflow (McDonnell and Beven, 2014). In order to investigate the flow pathways of rainwater through the snowpack, as well as the source contributions of snowpack outflow and rainwater to streamflow, we used stable water isotopes as environmental tracers.





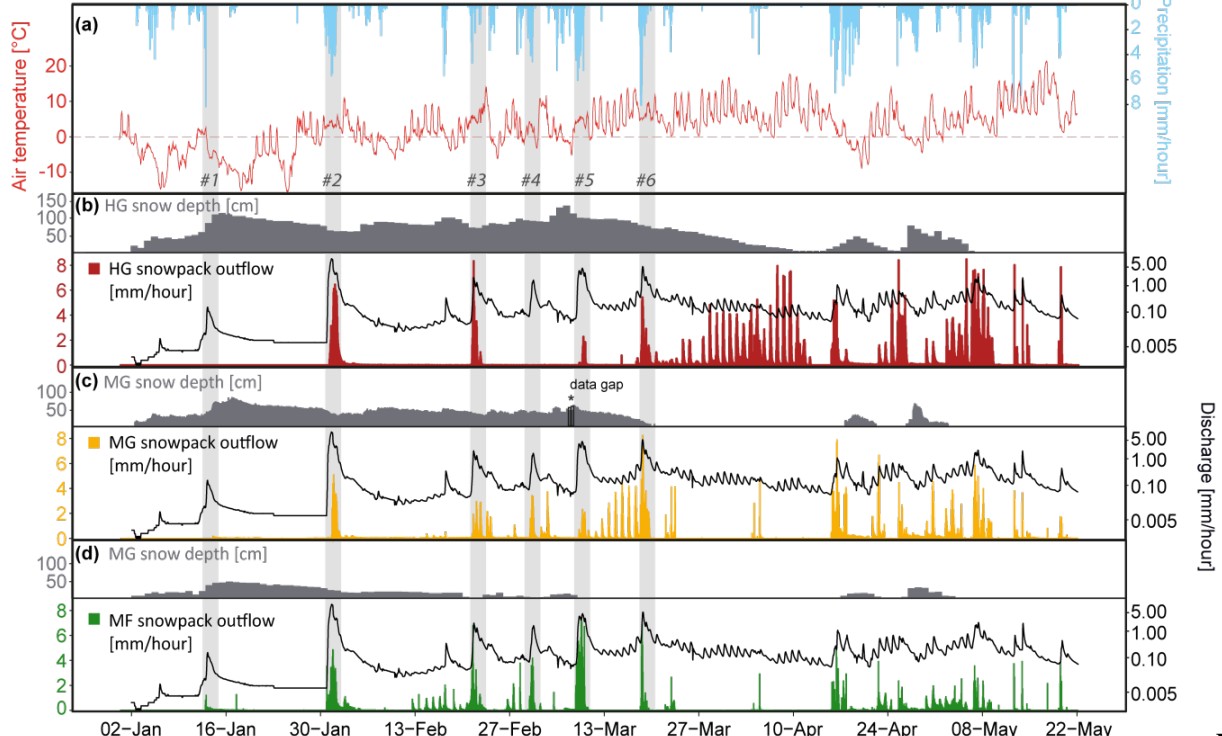

**Figure 5:**
**Hourly measurements of (a) precipitation (snow- and rainfall; blue), snow depth (grey) and air temperature (red) measured at the MG site. Discharge at the Erlenbach catchment (log-scale, panels b, c, d) and snowpack outflow at (b) high-elevation grassland site (HG, red), (c) mid-elevation grassland site (MG, yellow), (d) mid-elevation forest site (MF, green) during the period 01 January - 22 May 2017. Vertical grey bars indicate six rain-on-snow (ROS) events that are analysed in this study. Due to a data gap in the snow depth sensor (* in panel c), daily snow depth values of the webcam are shown for that period.**

## 3.2 Contribution of snowpack outflow and rainfall to catchment outflow

### 3.2.1 Isotopic composition of rainwater and snowpack outflow

Figure 6 compares the isotopic composition of water samples collected with the three lysimeter systems during the 2017 winter period. Because the lysimeter funnels were permanently installed, they collected snowmelt and rain-on-snow during snow-covered periods, as well as rainfall during snow-free conditions. We thus classified the samples either as rain, rain-on-snow or snowmelt to better quantify the effects of elevation and vegetation cover on the isotopic signatures of the different water sources. Because of the more persistent snow cover at the HG site, rainfall occurred only as rain-on-snow during the study period, so the HG lysimeter system collected predominantly snowmelt or a mixture of rain and snowmelt. Additionally, the isotopic composition of bulk snow samples at the HG and MG sites are shown (no regular bulk snow sampling was carried out at the MF site).





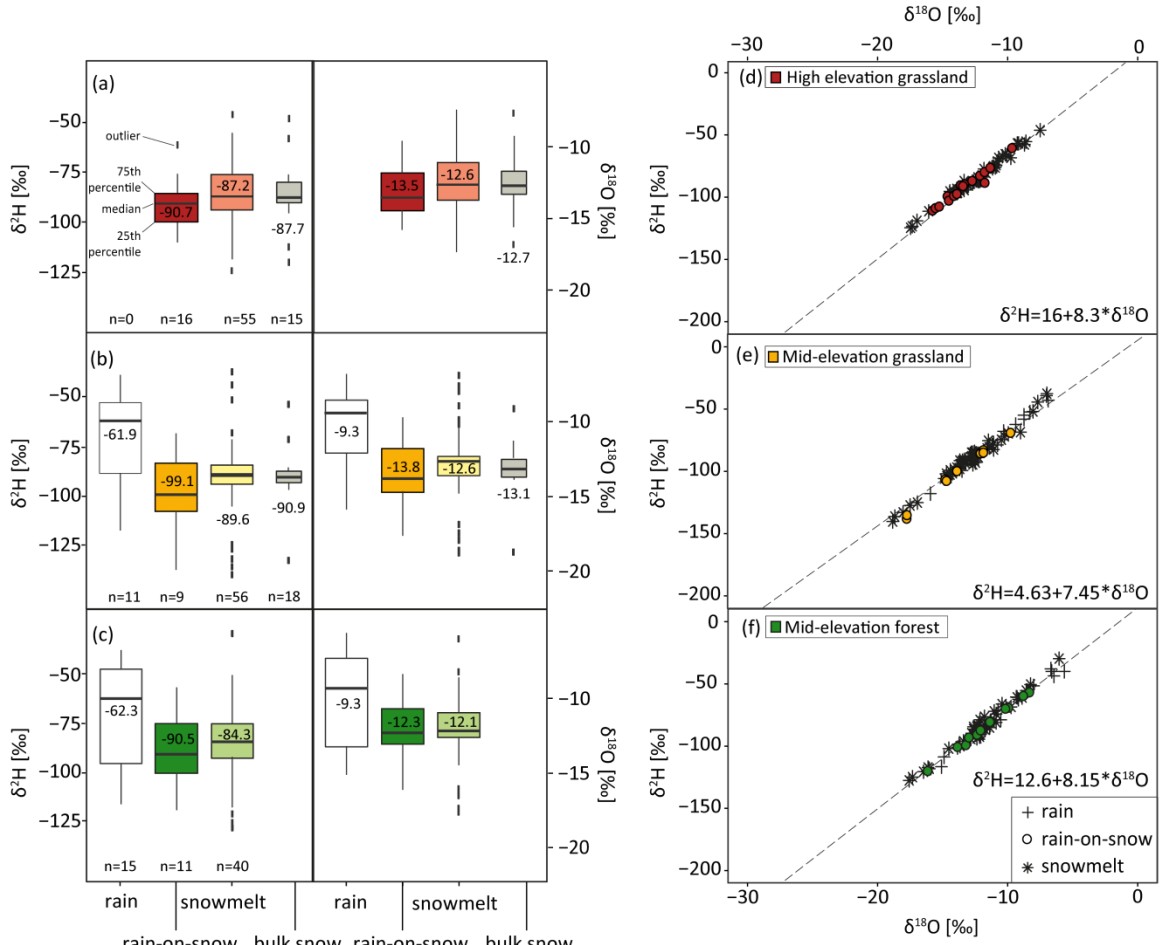

**Figure 6: Isotopic composition of rainwater, rain-on-snow, snowmelt and bulk snow measured at the three lysimeter sites (a, d) high-elevation grassland (HG) site, (b, e) mid-elevation grassland (MG) site and (c, f) mid-elevation forest site for the study period 1 January – 5 May 2017. Except for bulk snow, all samples were collected with the same lysimeter systems; rainwater was collected during snow-free conditions, whereas rain-on-snow and snowmelt were collected during conditions with snow-cover. The isotopic composition of bulk snow is added for the HG and MG site (panels a and b; grey). Panels (a-c) show boxplots of the isotope values $\delta^2H$ (left) and $\delta^{18}O$ (right). Panels (d-f) show the isotope values $\delta^2H$ and $\delta^{18}O$ plotted in dual isotope space together with the local meteoric water lines (dashed lines and equations, derived from rainwater samples collected at each field site between May and October in 2016 and 2017).**

Rainwater at the MG site and throughfall at the MF site had similar isotopic compositions (differences in median $\delta^2H$ and $\delta^{18}O$ were 0.4 ‰ and 0.0 ‰, respectively), however, interception and mixing of rainwater in the forest canopy resulted in a wider range of isotope values in throughfall at the MF site compared to the grassland site (Figure 6b, c). Our data show further that rain-on-snow and snowmelt under forest canopy (MF site) were isotopically heavier than the corresponding samples from the nearby grassland (MG) site. The absolute differences in the median $\delta^2H$ values between the two sites were 8.6 ‰ and 5.3 ‰ for rain-on-snow and snowmelt, respectively (the corresponding differences in median $\delta^{18}O$ were 1.5 ‰





and 0.5 ‰ for rain-on-snow and snowmelt, respectively). This isotopic difference suggests that canopy-intercepted snow at the forest site underwent enhanced isotopic fractionation such that throughfall (and thus the snowpack) became isotopically heavier under forest cover compared to open grassland (Claassen and Downey, 1995; Koeniger et al., 2008).

At the 200-meter higher grassland (HG) site, the median isotopic composition of bulk snowpack and snowpack outflow (rain-on-snow and melt) was heavier than at the lower grassland (MG) site; the median $\delta^2$H values differed by 8.4 ‰ in rain-on-snow, by 2.4 ‰ in snowmelt and by 3.2 ‰ in bulk snow (Figure 6a, d), but the isotopic differences in median $\delta^{18}$O were smaller than 0.4 ‰. The isotopically heavier bulk snow and snowpack outflow at the higher site is the opposite of the expected altitude effect (Dietermann and Weiler, 2013; Moser and Stichler, 1970), but one must remember that the snowpacks at the three different sites lasted for different spans of time, during which they received different rain and snow inputs with different isotopic compositions. One should therefore not expect to see a conventional altitude effect (which in any case would be small) in field data like ours.

### 3.2.2 Temporal and spatial isotopic variation of bulk snow and snowpack outflow during rain-on-snow events and snow melt

Due to frequent melt periods and ROS events, values of $\delta^2$H in bulk snow and snowpack outflow were highly variable over time at all three lysimeter sites (Figure 7). Similar to other studies (Gustafson et al., 2010; Taylor et al., 2002a), our data show that snowpack outflow, and thus catchment recharge, is much more variable in time than would be implied by weekly bulk snow samples alone. Nonetheless, bulk snow samples at the HG and MG sites mirrored the general isotopic pattern of the snowpack outflow samples. For instance, at the HG site, both sample types indicate a clear isotopic enrichment during melt-out of the seasonal snowpack in early April 2017 (Figure 7a). However, bulk snow samples, which were collected only weekly or twice a week, could not capture the high temporal variability that was observed in snowpack outflow (e.g. during ROS event #6 at both HG and MG sites, and during event #5 at the MG site; Figure 7).

Our daily isotope measurements of rainwater and snowpack outflow across the catchment landscape allow for studying the temporal and spatial isotopic variation of snowpack outflow during rain-on-snow events and snow melt. Our main observations are:

*1. Snowpack outflow reflects the isotopic composition of incoming precipitation*

During most ROS events, the isotopic composition of snowpack outflow reflected that of incoming rainfall (#'s 1, 3, 5, and 6; Figure 7; no rainfall data available for #2). For instance, during event #5, $\delta^2$H in incoming rainwater was -43.5 ‰ and $\delta^2$H in snowpack outflow (rain-on-snow) of the changed from -80.0 ‰ to -137.3 ‰ and from -88.8 ‰ to -110.5 ‰ at the MG and HG site, respectively.




2. *Snow depth controls isotopic response to ROS events*

Similar to the snowpack outflow volumes (Sect. 3.1.3), the isotopic response of the snowpack to individual ROS events likely depended on the local initial snowpack properties and the event magnitude. Isotopic responses in snowpack outflow were more damped at the HG site compared to the signals measured at the MG and MF sites, because the snowpack was
deeper at the higher elevation site. A similar effect of the snow depth was also apparent at all other sites: the isotopic variability of snowpack outflow was smaller when the seasonal snowpack was relatively deep (e.g., between events #2 and #5 at the MG site), and the variability increased when the snowpack became shallower, including during the two short-term snowpacks (e.g., between 17 April and 4 May 2017 at the MG site). At the MG site, rainwater and snowpack outflow had very similar isotopic compositions during event #6 (i.e., no damping), because the ripe shallow (17 cm) snowpack enabled
the vertical percolation of incoming rainwater (Figure 7b; Kroczynski, 2004). At the HG site, however, the snowpack was deeper (91 cm) during event #6, and incoming rainwater was mostly retained in the snowpack, resulting in a damped isotopic response in snowpack outflow (Figure 7a).

The isotopic signal of incoming rainwater can be altered as it percolates through the snowpack, depending on snow metamorphism and isotopic exchange (Judy et al., 1970). A significant isotopic depletion or enrichment of snowpack
outflow due to such rain-on-snow events has already been reported in other studies (Herrmann, 1978; Juras et al., 2016; Shanley et al., 1995; Unnikrishna et al., 2002). The isotopic exchange in the snowpack is mainly controlled by the residence time of liquid water (snowmelt and rain-on-snow) in the snowpack, which, in turn, is determined by the depth and the density of the snowpack (Taylor et al., 2001; Taylor et al., 2002), the rainfall magnitude (Herrmann, 1981), and the flow rate of percolating liquid water. As a result, deeper snowpacks generally cause a slower rainwater throughflow, which enhances
isotopic redistribution in the snowpack, and isotopic exchange between the liquid water and solid ice (Lee et al., 2010; Lee et al., 2010b; Taylor et al., 2001).





### 3. *Light isotopes preferentially leave the snowpack during melt*

Isotopic variations in snowpack outflow can also result from freeze-melt processes in the snowpack during rain-free periods. For instance, snowpack outflow at the MG site became isotopically lighter than bulk snow after ROS event #3, despite rainwater being isotopically heavier than bulk snow during ROS event #3 (Figure 7b).

The isotopic contrast between the snowpack outflow of the last day of event #3 and the following day was 13.4 ‰ for $\delta^2H$ and 2.2 ‰ for $\delta^{18}O$. This depletion signal occurred simultaneously with a decrease in air temperature to below 0 °C, suggesting isotopic fractionation effects in the snowpack because of partial phase transitions of liquid water to ice (Herrmann et al., 1981; Shanley et al., 1995; Stichler et al., 1981; Taylor et al., 2001). During partial freezing, the liquid phase becomes isotopically lighter, because the heavier isotopes preferentially transition into the solid phase, i.e., the lower free energy state (Hoefs, 2018). During this partial freezing process, lighter isotopes preferentially leave the snowpack, and over cycles of melting and refreezing, the snowpack becomes isotopically heavier and more homogeneous (Huth et al., 2004; Judy et al., 1970; Lee et al., 2010b; Schmieder et al., 2016; Taylor et al., 2002b, 2001; Unnikrishna et al., 2002). During the melt-out period of the seasonal snowpack, this fractionation effect results in snowpacks and snowpack outflows that become isotopically heavier over time. This trend can be observed at the HG site, where rising air temperatures and dry conditions between 26 March and 9 April 2017 resulted in progressive melt of the seasonal snowpack (Figure 7a). This melt-out of the seasonal snowpack was accompanied by a gradual isotopic enrichment in snowpack outflow $\delta^2H$ from -96.5 ‰ to -84 ‰. The isotopic composition of streamflow mirrored this isotopic trend in snowpack outflow, suggesting that snowmelt from higher elevations contributed to catchment outflow during the melt-out period.







Figure 7: Deuterium ($\delta^2$H) concentrations in precipitation (snow- and rainfall; light blue) and snowpack outflow (separated in rain, rain-on-snow and snowmelt) indicate spatial and temporal variability represented by the (a) high-elevation grassland (HG; red) site, (b) mid-elevation grassland (MG; yellow) site and mid-elevation forest (MF; green) site (d) and in stream water (grey) at the Erlenbach outlet during the study period 01 January - 22 May 2017. Stream water isotopic composition (grey) is shown in panels (a)-(c) for comparison (1 January - 14 May 2017) including dashed lines representing the range between -85 ‰ and -75 ‰. Error bars indicate the standard error of the isotopic composition of snowpack outflow due to spatial heterogeneity at the plot scale.

### 3.2.3    The contribution of rainfall and snowpack outflow to river discharge

*1.  Implications of isotopic variability in snowpack outflow for end member mixing analysis*

Figure 7 shows that the isotopic signal of Erlenbach stream water responded to the rainfall and snowpack outflow during the individual ROS events. In the following section we quantify the contribution of rainfall and snowpack outflow to streamflow, using stable water isotopes as conservative tracers in two-component hydrograph separations. These analyses were carried out individually for each sampling site using the volumes and isotopic compositions of their snowpack outflows. Thus, our results reflect the relative snowpack outflow contribution to streamflow for three different scenarios that assume that the catchment-average snowpack is represented by the mid-elevation grassland (MG), mid-elevation forest (MF) or high-elevation grassland (HG) site, respectively. For comparison, we also performed hydrograph separation using rainfall as the "new water" end member. In all cases, pre-event stream water isotopic composition was used as the "old water" end member, following conventional practice in two-component hydrograph separations. Here we present our results based on $\delta^2$H, for which the temporal variations in stream water were larger, and the measurement uncertainties were smaller, compared to $\delta^{18}$O.

*2.  IHS results are highly variable across sites and rain-on-snow (ROS) events*

Figure 8 summarizes the estimated contributions of rainfall and snowpack outflow to peak streamflow during the six ROS events. Snowpack outflow contributions to streamflow varied among the six ROS events and the three snowmelt lysimeter sites, ranging from 34±7 % to 42±2 % at the HG site, from 13±1 % to 58±3 % at the MG site, and from 7±4 % to 91±20 % at the MF site (Figure 8, Table 3). The different results among the three sampling locations reflect the highly variable isotopic compositions of the snowpack outflow across the catchment. For example, during event #6, isotope data from the HG site suggested a significant snowpack outflow contribution to discharge (34±7 %), whereas isotope measurements from the lower-elevation MG site implied a much smaller (13±1 %) contribution of snowpack outflow to streamflow. The different estimates at the two sites can be explained by the stronger retention of incoming rainfall in the higher-elevation (HG) snowpack (resulting in snowpack outflow that was isotopically closer to streamflow) and the





transmission of rainfall through the snowpack at the MG site (resulting in snowpack outflow that resembled the isotopically light incoming precipitation; Sect. 3.1.3; Figure 8; Table 3).

This comparison raises an important point of interpretation. Any two-component hydrograph separation is based on the fundamental assumption that there are only two end members (in our case, one of the snowpack outflows, and "old water" represented by pre-event streamflow). Thus one cannot interpret the results above as demonstrating that more snowpack outflow reached the stream from high elevation sites like HG than from lower elevation sites like MG. Instead, what these results show is that if the catchment-wide snowpack outflow resembled that from the MG site, it could only make a small contribution to streamflow (because otherwise the peak streamflow would need to be isotopically lighter than it in fact was), but that if the catchment-wide snowpack outflow resembled that from the high-elevation HG site, it could plausibly make a larger contribution to streamflow.

### 3. *Larger contribution of snowpack outflow to streamflow compared to rainwater*

The contributions of rainfall to streamflow during four events (#1, #3, #5 and #6) ranged between 5±2 % and 34±2 % (no estimate could be obtained for event #2 due to a gap in rainwater isotope sampling, and unrealistic hydrograph separation results were obtained for event #4). Based on snowmelt lysimeter data from the MG site (unrealistic hydrograph separation results were obtained for event #3), the contributions of snowpack outflow to streamflow were larger than those of incoming rainfall during three events (34±2 % vs. 58±3 % for event #1, 25±1 % vs. 50±5 % for event #5, and 12±1 % vs. 13±1 % for event #6; Table 3). The isotopic composition of snowpack outflow at the MG site was often more damped compared to that of rainwater during most events because of mixing and fractionation processes in the snowpack (Sect. 3.2.2; Figure 8). As a consequence, the snowpack outflow was isotopically more similar to that of streamflow, which resulted in larger fractions $F_{spo}$ compared to $F_R$.

Although the number of ROS events in our data set is small, our results are in line with previous studies showing that the differences between hydrograph separation results obtained for rainwater and snowpack outflow can potentially be large and should be considered in snow-dominated catchments (Buttle et al., 1995). Our analysis assumes that the end members of the different scenarios (i.e., snowpack outflow at the MG, MF or HG site) are representative for the whole Erlenbach catchment. However, the catchment is characterized by a diverse vegetation cover (22 % partially forested, 53 % forested and 25 % grassland) and surface topography (altitude 1000-1500 m asl), so that the "real" contribution of snowpack outflow to streamflow is likely to lie between the estimates derived from the three scenarios. The hydrograph separation estimates for the three lysimeter sites can only provide a probable range of snowpack outflow contributions to discharge from different landscapes of the catchment. As shown here, the estimated contributions of snowpack outflow to streamflow can vary considerably due to differences in landscape characteristics, rainfall magnitude and snowmelt processes. Future sampling strategies should take this spatial and temporal variability in snowpack outflow into account.





**Table 3: Contributions of rainfall or snowpack outflow to daily peak discharge based on two-component isotope hydrograph separation (IHS) using δ²H. The IHS was carried out with four different isotope data sets that were collected with snowmelt lysimeters at the HG (high-elevation grassland) site, MG (mid-elevation grassland) site and MF (mid-elevation forest) site and with the rainfall collector at the catchment outlet.**

| ROS event number | Fraction of daily peak discharge ±SE (-) | | | |
|---|---|---|---|---|
| | Snowpack outflow HG | Snowpack outflow MG | Snowpack outflow MF | Rainfall MG |
| #1 | a) | 0.58 ± 0.03 | 0.76 ± 0.30 | 0.34 ± 0.02 |
| #2 | 0.42 ± 0.02 | b) | 0.91 ± 0.20 | b) |
| #3 | -0.16 ± 0.08 | -1.28 ± 1.43 | 0.07 ± 0.04 | 0.05 ± 0.02 |
| #4 | a) | 0.29 ± 0.04 | 0.46 ± 0.06 | 2.28 ± 2.13 |
| #5 | 2.63 ± 0.64 | 0.50 ± 0.05 | 0.32 ± 0.02 | 0.25 ± 0.01 |
| #6 | 0.34 ± 0.07 | 0.13 ± 0.01 | 0.20 ± 0.01 | 0.12 ± 0.01 |

a) no snowpack outflow occurred

b) data gap

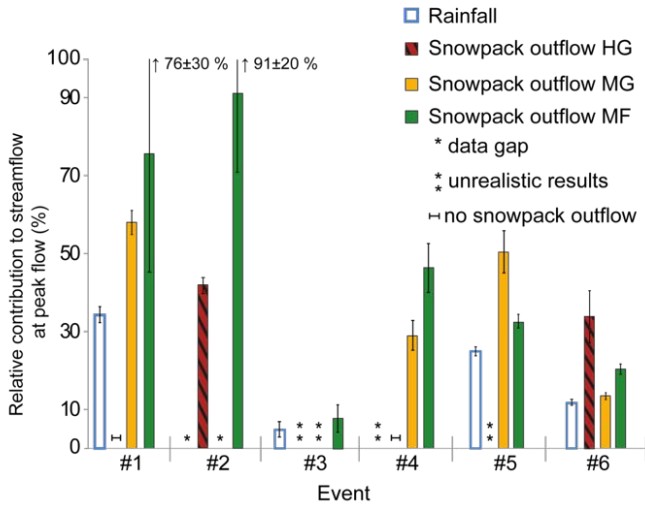

**Figure 8: Relative contribution of snowpack outflow to streamflow at peak flow based on isotopic hydrograph separation for the six rain-on-snow events from winter 2017, including the incoming rainfall (blue, not filled) and the snowpack outflow of the high-elevation site (red, black-shaded), mid-elevation site, grassland (yellow) and the mid-elevation site, forest (green). For some events, no data (*) were available (no melt) or the results were unrealistic (**). The error bars indicate the standard error of the snowpack outflow contribution to streamflow (see section 2.4).**



## 4        Summary and conclusions

In Switzerland, rising air temperatures are likely to lead to more frequent rain-on-snow (ROS) events in the future, and thus to increased risks of destructive floods, partly due to enhanced snowmelt. However, the processes leading to such enhanced melt are spatiotemporally heterogeneous, and predicting streamflow peaks induced by ROS events requires a better

understanding of how water sources contribute to streamflow.

Using three automated snowmelt lysimeter systems, located along an elevation gradient of 1185 to 1420 m asl in a partly forested pre-Alpine catchment, we were able to capture the spatial and temporal variability of snowpack outflow generated over the winter season (Figure 2; Figure 5). A comparison of snowpack properties at a grassland and a nearby forested site showed that canopy interception significantly reduced incoming snowfall, and thus the maximum snow depth under forest

cover was around 20 cm shallower compared to open grassland. Measurements from two grassland lysimeter sites located at different altitudes (1220 and 1420 m asl) showed that the snowpack was on average 55 cm deeper, and snowmelt occurred 21 days later, at the 200 m higher site.

To better understand how snowpack outflow is generated during ROS events across the catchment landscape, we studied ten ROS events in greater detail (Figure 3). We found that the snowpack outflow volumes during ROS varied considerably

across the three lysimeter sites, and that this variability was linked to rainfall characteristics and initial snowpack properties (Figure 4). Initial snow depth and rainfall volume explained most of the event-to-event variability in snowpack outflow volumes. Overall, more rainwater was retained in the snowpack at the grassland sites (Figure 3), which had deeper snowpacks compared to the forest site. Our data show that long and high-intensity ROS events can result in particularly high discharge peaks, even in mid-winter when the snowpack is not saturated (e.g., ROS event #2). This suggests that enhanced

snowmelt during ROS events, and/or high antecedent moisture due to ongoing snowmelt, are not limited to late winter when the snowpack is mature and saturated.

We used daily stable water isotope measurements in snowpack outflow, rainwater and stream water to draw inferences about transport and mixing of rainfall within the snowpack during individual ROS events. Depending on the local rainfall characteristics and the snowpack properties, the isotopic responses in snowpack outflow could be either strongly or weakly

damped, indicating large spatiotemporal variations of the snowmelt process (Figure 7). Consequentially, isotope-based two-component hydrograph separation (IHS) for estimating snowpack contributions to streamflow often yielded very different results (Figure 8), depending on which site-specific snowpack outflow isotopic compositions were used. This range of IHS results provides reasonable estimates of relative snowpack outflow contributions to streamflow during individual ROS events, under the assumption that the three lysimeter sites are representative for the snowmelt processes at the catchment

scale. Further, the range of our IHS results indicate that in steep, partly forested catchments, estimates of snowpack outflow contributions to streamflow derived from bulk snow samples or outflow samples collected at only one location can be highly uncertain. This is in line with Fischer et al.'s (2017) study that showed strong spatial variability in rainwater isotopic




composition in the southern Alptal catchment. Using rainwater isotope data in the IHS analysis suggests that the relative

contribution of rainwater to streamflow may often be much smaller than the contribution of snowpack outflow, because

snowpack outflow is a mixture of both rainwater and snowmelt. Our analysis suggests that snowpack outflow can contribute

substantially to streamflow during ROS events and that these contributions depend strongly on the local snowpack properties

and rainfall characteristics.

In order to obtain more realistic estimates of snowpack outflow contributions to streamflow during ROS events,

snowpack outflow volumes and their isotopic compositions could be interpolated across the study area using a spatially-

distributed snowmelt model. Recent snowmelt modelling approaches at the catchment scale do not, however, explicitly

simulate snowpack outflow during rain-on-snow events (Ala-aho et al., 2017; Lyon et al., 2010; Smith et al., 2016), or use

stable water isotopes to track the flow pathways (Kormos et al., 2014; Marks et al., 2001; Rössler et al., 2014; Storck et al.,

1998). Our spatiotemporally distributed isotope measurements could thus be beneficial for testing and improving existing

snowmelt models (Zappa et al., 2015).

**Acknowledgements**

This project was supported by the Swiss National Science Foundation SNF through the Joint Research Projects

(SCOPES) Action (Grant IZ73Z0_152506). We thank the staff of the Swiss Federal Institute for Forest, Snow and

Landscape Research (WSL), especially Massimiliano Zappa for his input on the manuscript, Karl Steiner for great support in

the field and Alessandro Schlumpf for isotope analysis. We also thank Bjørn Studer and Daniel Meyer for the help in the

laboratory, as well as the Oberallmeind Schwyz and Martin Brun who authorized the installation of one snowmelt lysimeter

system on their land.

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
