# Peer review of "Monitoring snowpack outflow volumes and their isotopic composition to better understand streamflow generation during rain-on-snow events"

_Hydrology and Earth System Sciences, 2019_

## Referee Comment (RC1) · Anonymous Referee #1 · 14 Mar 2019

General comments: The study of Rücker et al. presents an analysis of rain-on-snow events in a Swiss Pre-Alpine catchment for two winters. The focus of this study lies on characterizing the snow conditions at the lysimeter measuring sites and analysing the snowmelt response with respect to the discharge response. Moreover, an isotope-based hydrograph separation provides estimates for rainfall vs. snowmelt contributions to stream runoff. In this context, the present work contributes to a better process understanding of rain-on-snow events, being important for flood management and model calibration. The manuscript is well structured and the language quality is appropriate. Although the manuscript sections seem to be well balanced to me, surprisingly, I could not find research hypothesis at the end of the introduction section. These would make

the study much stronger and would directly lead to the titles already chosen in the result section. With respect to the result and discussion section, I noticed that the discussion part is sometimes too short and references could be added. This lack probably results from merging result and discussion section. A last point addresses the use of tenses. I recommend to the authors to check again the when present tense and past tense was used. For example, results are normally described in past tense, which sometimes is not the case.

Specific comment: Page 3, Line 11: provide a reference for the effect of the canopy structure Page 3, Line 21: please comment whether sublimation plays a role in this context as well Page 3, Line 29: snowmelt contribution Page 4, Line 1-4: please rephrase; how do you justify the rain snow transition zone? Page 4, Line 8: add more details, such as elevation of these measuring stations Page 5, Line 7: please argue on the representativeness of your lysimeter sites with respect to the catchment (aspect, slope, elevation). What was the reason behind selecting MG and MF sites so close to each other? Page 5, Line 13: 30 m difference in elevation is redundant with line 8 Page 6, Line 13: please specify the improvements made for this site Page 6, Line 23: please comment and add in the text on fractionation though evaporation? Could the sampling bottles automatically closed after filling? Page 7, Line 24: did you use a recognition software to transfer webcam pictures into snow depth data? Page 7, Line 25: was HG site subject to blowing snow? Page 8, Line 3: please use references to support these criteria Page 8, Line 15: were collected Page 10, Line 7: please use a reference for the Gaussian error propagation Page 11, Line 11: please characterize these cold conditions, how was the mean air temperature? Page 12, Line 6: replace "several times" by a number to better quantify Page 14, Line 6: replace by "Further four ROS" Page 15, Line 15: provide some statistics when reporting statistical significance Page 15, Line 21: 200-meter is already known and thus redundant, please remove Page 16, Line 9: provide some statistics when reporting no statistical significance Page 17, Line 8: to which processes do you refer to? Please rephrase. Page 17, Line 21: provide some statistics when reporting no statistical significance Page 20, Line 4: be more quantitative with respect to variable responses and lag times Page 23, Line 4: 200-meter is already known, please remove Page 23, Line 10-11: this is not clear. Is the elevation gradient defined by your sites not large enough to show the elevation effect? Page 23, Line 24: provide results from a statistical test to show the similarity in isotopic composition Page 30, Line 6: replace by "By using" Page 30, Line 10: replace by "compared to that of open grassland" Page 30, Line 18: as this is the summary, it would be helpful to repeat the initial criteria how you defined the ROS events (precipitation amount, initial snow depth threshold) Page 30, Line 26: IHS is already introduced before Page 31, Line 23: the correct co-author name is McNamara

Fig. 1: add a map of Switzerland locating your study site. Why is the forest in the lower part of the map dark green? Shading effect of the underlain hillshade? (in this case, hillshade data not present in the legend) MG seems to lie in the forest.

Fig. 2: please make air temperature line thicker and improve grey bars, which are not so visible

Fig. 5: The snow depth subplots could be taller to increase visibility

Fig. 7: what is the meaning of the grey dashed line in all subplots?

Fig. 8: why are event #3 and #4 results unrealistic?

---

## Referee Comment (RC2) · Roman Juras (Referee) · 23 Apr 2019

**General comments:**

The paper presents very interesting and current topic about snowpack outflow contribution to the catchment outflow during rain-on-snow (ROS) events. The authors identified ten ROS events during two winter seasons, where the effect of snow cover and further snowpack outflow to the stream were analysed. Authors employed two-component hydrograph separation method using natural stable water isotopes and enhanced system of water sampling. I like the study very much, because understanding of the hydrological processes during ROS is still not sufficient and this study aims to contribute to this knowledge. It is an interesting study and worth to publish in HESS. Nevertheless, I recommend to do some minor revisions and I also have a couple of suggestions to improve the study.

My major point to the study is that the authors should present results from the hydrograph separation and provide more information about snowpack outflow composition. Since the isotopic content of rain and snowmelts during ROS events were sampled, the rainwater contribution in the outflow can be easily calculated. The authors can also provide the separated hydrograph with all the components.

The authors define in section 2.2.1 the ROS event. Maybe I just missed something, but from this definition it seems that duration of ROS equals duration of rain. This does not match with the values in Tab. 1 (see columns *Start time, End time* and *Rainfall duration*). This issue is also connected with total ROS outflow volumes. Please describe it clearer.

The *Introduction* section usually provides in the end some basic goals of the paper. I miss this part in the particular section. Please reformulate the last paragraph (Page 3, lines 30 – 33, Page 4, lines 1 – 3).

Although, I am not a native english speaker, I recommend some proof reading regarding the language.

**Specific comments:**

- Please use elevation units as "m a.s.l." and not "m asl".

- Please present what time zone do you use (UTC, CET, etc.).

- Figure1: Can you add an information about coordinate system of the map and

how far is HG site from the catchment. You should also add a small map of Switzerland, where the study site is located.

- Lot of technical information regarding the field monitoring system is provided (page 5, lines 15 – 18, page 1 – 5). It would be beneficial to better readers clarity if you present these information in tabular form. The sketch or photograph of the monitoring system would be also very practical and provide better view how the system works.

- If you state just water stable isotopes as such, do not use $\delta$ symbol, but only $^2$H and $^{18}$O (Page 3, line 3). Delta symbol refers to some defined standards.

- You mention that the snow was sampled by a snow tube (Page 7, lines 27 – 28). Do you use any standardised tube? What is the material of the tube?

- Please be consistent with presenting the time intervals. You often mix numbers and text information, like 10-minute x ten minute (i.e. Page 7, lines 7, 9).

- Page 16, line 1: According to Fig.2 the snow depth at HG does not look 97 cm deeper than MG.

- Page 16, line 14 – 15: Do you have any isotopic signature results of the throughtfall? Can you compare it with rainfall on the open sites?

- Page 18, line 2: How did you estimate the cold content of the snow? Did you also measure the snow temperature or did you just guess it from the air temperature? If you consider just mean air temperature, how long prior to the event? Maybe you should rather use cumulative temperature from last x hour. Nevertheless, this statement is quite tricky, because the higher cold content does not always mean that more incoming rainwater is stored in the snowpack. Water storage is more related to the snow stratigraphy and layering.

- Figure 4: Can you add $r^2$ values to all subplots?

- Page 20, line 4: How do you define lag times?

- Page 20, line 6: How do you estimate saturation of the snowpack?

- Table 3: What does represent the last column (Rainfall MG) of the table? There are used two terms in figure 6 – *rain* and *rain-on-snow*. Please be consistent with the naming. There are presented results of different water contribution to the catchment outflow only during peak discharge. Can you also present results of outflow composition from the entire event period?

---

## Referee Comment (RC3) · Daniele Penna (Referee) · 29 Apr 2019

General comment

First of all, I apologize with the Authors and Editor for my late review.

This is a very interesting manuscript that focuses on the role played by rain-on-snow (ROS) events in enhancing snowpack outflow and thus snowmelt, ultimately contributing to stream runoff. I worked for some years in a snow-dominated catchment and I had the opportunity to observe the significant impact that ROS events have on the catchment hydrological response in the melting period. Therefore, experimental work that

provides a better understanding of the controls on snowmelt contribution to streamflow during ROS events is welcome and certainly appreciated by the readers of HESS.

The manuscript is well written, solidly structured, nicely illustrated, with updated and relevant references, and the data well support the results interpretation. I basically agree with the comments by the two other Reviewers and I overall like the response of the Authors. I have only a few specific comments that I hope can contribute to improving the manuscript. In the end, I recommend a minor revision before publication.

Specific comments

- In agreement with Reviewer 1, I also noticed the lack of a clear and testable research hypothesis stemming from the knowledge gaps defined earlier in the Introduction. The Authors replied that the main hypothesis ". . .is that vegetation and elevation substantially affect the generation and the isotopic composition of snowpack outflow, and thus snowmelt contribution to streamflow. In my opinion, this reply is not fully satisfactorily. First, "vegetation" is quite a vague term in this context: reading the rest of the manuscript and knowing the area it is clear that this term refers to forest trees but, in principle, this could be valid for understory vegetation as well. So, I suggest being more specific here. Secondly, what does it mean that vegetation and elevation affect snowpack outflow generation? I guess the Authors mean outflow amount or volumes, but again this should be specified. Most importantly, this is only the general hypothesis. I suggest to complement it with some specific hypotheses or specific research questions that better address the core of this work and around which the Results and Discussion section could be built. For instance, one specific research question could focus on the role of rainfall characteristics and initial snowpack properties on the variability of snowpack outflow volume. Another specific research question could deal with the spatial variability of snowmelt contribution to streamflow in the catchment (comparison of hydrograph separation results among the three sites) and a third one to the temporal variability of snowmelt contribution to streamflow (comparison of hydrograph separation results among different ROS events). These are only suggestions but I think

that structuring the Results section so that its parts reflect the specific questions posed at the beginning would tell a clearer story and accompany the reader in a more linear way.

- I think that what would be really interesting and novel is the application of three-component hydrograph separation to quantify the proportion of rainfall and pre-event snowmelt during ROS events. As far as I understood, the instrumental design and the sampling scheme would allow for the application of this mixing model that, of course, requires the availability of a second tracer. Is there any additional tracer available? Is the application feasible? Are there theoretical or practical constraints that prevent this analysis? I wonder if the Authors already planned a follow up of this study considering this aspect. A comment on this is welcome.

- At lines 268 and 269 the Authors stated that the pre-event tracer signature (by the way, talking about isotopes I think that the terms "signature" and "composition" are more appropriate than "concentration") was determined by sampling the stream on the day prior the ROS event. In a previous study in a snowmelt-dominated Alpine catchment (Penna et al., 2016, JoH, https://doi.org/10.1016/j.jhydrol.2016.03.040), we compared two different methods to determine the pre-event stream signature for two-component hydrograph separation during snowmelt, ie the average of several samples taken during baseflow and a sample taken before the snowmelt-induced runoff event. We found, in some cases, marked differences in the estimated snowmelt proportions in the stream using the two methods, and we related these differences to the fact that streamflow may have contained a small amount of residual snowmelt water at night, especially late in the melt season, so that meltwater influenced the isotopic composition of the stream between melt events. In the case presented by the Authors, the ROS events occurred in winter (Jan-March) and so this effect might not be so important but, nevertheless, I wonder if this effect could happen here as well at least in the late winter events (eg, March). A comment on this could be useful.

Minor comments and technical corrections L126. Remove "at mid elevations".

L147-148. I suggest shortening the title.

L179: What is the relative measurement uncertainty of the tipping bucket?

L223. Remove the delta sign, it's not needed here.

L255. Replace "concentration" with "composition".

L259-260. Did the Authors/technicians apply any procedure to mitigate the carry over (memory) effect that can affect laser isotopic measurements when analysing subsequent samples with much different isotopic composition?

Fig. 3b: Add "rainfall" before the word "retention" below the 1.1 line.

L445-446. I suggest skipping this, redundant with what previously mentioned in the M&M section.

L456. Is the regression statistically significant? Can the Authors report the p-value of this regression?

L540-542. This sentence is not necessary and can be skipped.

Fig. 6. It is not immediately clear to me which rain samples are, which snowmelt samples, and ROS samples and bulk snow, so I suggest making the box plot clearer. In addition, did the Authors perform a statistical analysis in order to check for the differences in isotopic composition? Moreover, I wonder if the slope of the regression lines in the dual isotope space (Fig. 6d-f) is statistically different between the MG site and the HG and MF sites (see, for examples, another isotopic study on rain and snowmelt in the Alpine catchment mentioned above, Penna et al., 2017, HP, https://doi.org/10.1002/hyp.11050). This could be performed and discussed in the light of the inter-site comparison in rain and snowmelt isotopic composition.

L658. Replace "concentration" with "composition".

L706-707. Which assumptions were violated to have unrealistic results?

---

## Author Response (AR1)

Swiss Federal Institute WSL   Mountain Hydrology and
Zürcherstrasse 111   Mass Movements
8903 Birmensdorf   Hydrological Forecasts
Switzerland   Andrea Rücker
Phone +41 (0)44 739 2111   PhD student
Fax +41 (0)44 739 2215   Phone direct +41 (0)44 739 2487
www.wsl.ch   or +41 (0)44 632 9172
andrea.ruecker@wsl.ch

[Figure]

Editors of Hydrology and Earth System Sciences

Zürich, May 14, 2019

**Manuscript re-submission**

"Monitoring snowpack outflow volumes and their isotopic composition to better understand streamflow generation during rain-on-snow events"

Dear Dr. Bettina Schaefli,

Please find attached the revised version of our manuscript entitled "Monitoring snowpack outflow volumes and their isotopic composition to better understand streamflow generation during rain-on-snow events" (hess-2019-11). We addressed all issues raised by the three reviewers and you can find our detailed responses and the track-changed manuscript and the supplement below.

We highly appreciated the thoughtful comments of the three reviewers, which helped to improve the manuscript. We also thank you for a timely handling of the manuscript and we are looking forward to the publication of our work.

We also found a glitch in our manuscript in Section 3.2.2, where we accidentally reported the wrong ROS number in event #6. Due to malfunctioning of the snow depth sensor, we adapted the snow depth in event #5. This improved the $R^2$ value in Figure 4.

The authors confirm that the article contains only original data. If you have any further questions please do not hesitate to contact me.

Thank you very much.

Sincerely yours, *Andrea Rücker*

**Response to reviewer 1**

**I thank the Anonymous Referee #1 for his comments. I have reproduced those comments below (in normal type), with my responses (in bold).**

General comments: The study of Rücker et al. presents an analysis of rain-on-snow events in a Swiss Pre-Alpine catchment for two winters. The focus of this study lies on characterizing the snow conditions at the lysimeter measuring sites and analysing the snowmelt response with respect to the discharge response. Moreover, an isotope based hydrograph separation provides estimates for rainfall vs. snowmelt contributions to stream runoff. In this context, the present work contributes to a better process understanding of rain-on-snow events, being important for flood management and model calibration. The manuscript is well structured and the language quality is appropriate.

**Thanks for these remarks.**

Although the manuscript sections seem to be well balanced to me, surprisingly, I could not find research hypothesis at the end of the introduction section. These would make the study much stronger and would directly lead to the titles already chosen in the result section.

**The fundamental hypothesis of our study is that vegetation and elevation substantially affect the generation and the isotopic composition of snowpack outflow, and thus snowmelt contribution to streamflow. We will include this statement in the revised version of the manuscript.**

With respect to the result and discussion section, I noticed that the discussion part is sometimes too short and references could be added. This lack probably results from merging result and discussion section.

**We are not sure what specific parts of the results and the discussion section the reviewer refers to. In any case, some of the processes discussed here (e.g., evaporation, sublimation, refreezing) have not been specifically investigated in our study, and thus we cannot go into further detail regarding individual mechanisms that cause changes in snowpack outflow or snowpack isotopic composition.**

A last point addresses the use of tenses. I recommend to the authors to check again the when present tense and past tense was used. For example, results are normally described in past tense, which sometimes is not the case.

**We will correct the use of tenses in the revised version of the manuscript.**

Specific comment:
Page 3, Line 11: provide a reference for the effect of the canopy structure.

**We will add the reference Koeniger et al., 2008.**

Page 3, Line 21: please comment whether sublimation plays a role in this context as well.

**We will include sublimation as a relevant process for isotopic enrichment of the snowpack in the revised manuscript.**

Page 3, Line 29: snowmelt contribution.

**We will change that in the revised manuscript.**

Page 4, Line 1-4: please rephrase;how do you justify the rain snow transition zone?

**We will indicate the exact elevation range covered with our snowmelt lysimeter systems. Within this elevation range, precipitation frequently shifts from snow to rainfall (Beniston, 2003; McCabe et al., 2007; Surfleet and Tullos, 2013; Zierl and Bugmann, 2005).**

Page 4, Line 8: add more details, such as elevation of these measuring stations

**We will add more details concerning the field sites.**

Page 5, Line 7: please argue on the representativeness of your lysimeter sites with respect to the catchment (aspect, slope,elevation). What was the reason behind selecting MG and MF sites so close to each other?

**The MG and MF sites were installed at official research plots where power supply was available (maintained by the Swiss Federal Institute for Forest, Snow and Landscape Research, WSL). The elevation, slope and aspect of the landscape are similar at the MG and MF sites, so that differences in snowpack outflow generation can be linked to the differences in vegetation cover. Because technical installations were not permitted in the upper part of the Erlenbach catchment, the HG site was installed at a nearby location with similar elevation and line power access.**

Page 5, Line 13: 30 m difference in elevation is redundant with line.

**We will change that.**

8 Page 6,Line 13: please specify the improvements made for this site.

**The improvements have already been described in detail in the next paragraph; we will add an introductory sentence to the first paragraph.**

Page 6, Line 23: please comment and add in the text on fractionation though evaporation? Could the sampling bottles automatically closed after filling?

**We will clarify the effect on fractionation on our sample bottles. The filled sampling bottles remained open until they were replaced with dry bottles once a week. One open sample bottle filled with 400 ml of a water sample of known isotopic composition was placed in the automatic water sampler each week to test whether evaporative fractionation occurred during the one-week sampling period. We did not find a substantial isotopic enrichment effect in these open sample bottles and thus assume that our sampling setup (automatic water sampler inside a protection hut) provides sufficient protection against evaporative fractionation of the snowpack outflow samples. For the automatic water samplers at the snowmelt lysimeter sites, the sample bottles were not modified. However, the sample bottles of the automatic watersampler sampling stream water of the Erlenbach were modified to reduce evaporative fractionation.**

Page 7, Line 24: did you use a recognition software to transfer webcam pictures into snow depth data?

**No, we did not use recognition software. We analysed the webcam pictures manually; we will clarify this in the manuscript.**

Page 7, Line 25: was HG site subject to blowing snow?

**We cannot exclude the possibility that wind drift occurred at the HG site, however, we did not observe substantial transport of snow due to wind during our field surveys. If wind drift occurred, it seems likely that wind drift affected all of the three individual lysimeter funnels similarly since they were located in close proximity to each other.**

Page 8, Line 3: please use references to support these criteria.

**These will be added.**

Page 8, Line 15: were collected.

**We will clarify this sentence.**

Page 10, Line 7: please use a reference for the Gaussian error propagation

**A reference to Genereux, 1998 will be added.**

Page 11, Line 11: please characterize these cold conditions, how was the mean air temperature?

**We will include the specific information: 22 cm of snow depth was reached during 6 days; mean air temperature was -6.6 °C.**

Page 12, Line 6: replace "several times" by a number to better quantify.

**This will be changed.**

Page 14, Line 6: replace by "Further four ROS".

**This will be changed.**

Page 15, Line 15: provide some statistics when reporting statistical significance

**We do not understand this comment of the reviewer as it refers to the caption of Figure 3: "(a) Rainfall 15 volumes at the MG site (light blue) and snowpack outflow volumes at the HG (red, black-shaded), MG (yellow) and MF (light green: winter 2017; dark green: winter 2018) sites."**

**If the reviewer refers to page 15, Line 10, we already provided a definition of statistically significant: "(i.e. larger than two times their pooled standard errors)." In the revised manuscript, we will provide the actual difference between the incoming rainfall and snowpack outflow volumes.**

Page 15, Line 21: 200-meter is already known and thus redundant, please remove.

**This will be removed.**

Page 16, Line 9: provide some statistics when reporting no statistical significance

**In the revised manuscript, we will provide the actual difference between the incoming rainfall and snowpack outflow volumes.**

Page 17, Line 8: to which processes do you refer to? Please rephrase.

**We will remove this part of the sentence.**

Page 17, Line 21: provide some statistics when reporting no statistical significance.

**We will add the p-value.**

Page 20, Line 4: be more quantitative with respect to variable responses and lag times.

**We will include more information about the ranges of snowpack outflow volumes and lag times between events and sites.**

Page 23, Line 4: 200-meter is already known, please remove.

**This will be removed.**

Page 23, Line 10-11: this is not clear. Is the elevation gradient defined by your sites not large enough to show the elevation effect?

**Correct, we argue that a significant effect of elevation on the isotopic composition in bulk snowpack cannot be seen at our field sites, probably because the elevation difference of 220 m was too small.**

Page 23,Line 24: provide results from a statistical test to show the similarity in isotopic composition.

**We think that the number of data points is too small to facilitate a statistical analysis. Instead, we will rephrase our observation and will say that the isotopic composition of snowpack outflow during ROS events responds to (not reflects) that of incoming rainwater.**

Page 30, Line 6: replace by "By using".

**This will be changed in the revised manuscript.**

Page 30, Line 10: replace by "compared to that of open grassland".

**This will be changed in the revised manuscript.**

Page 30, Line 18: as this is the summary, it would be helpful to repeat the initial criteria how you defined the ROS events (precipitation amount, initial snow depth threshold)

**This will be added in the summary.**

Page 30, Line 26: IHS is already introduced before.

**OK, but we should keep this abbreviation in the summary to improve readability.**

Page 31,Line 23: the correct co-author name is McNamara

**This will be corrected.**

Fig. 1: add a map of Switzerland locating your study site. Why is the forest in the lower part of the map dark green? Shading effect of the underlain hillshade? (in this case, hillshade data not present in the legend) MG seems to lie in the forest.

**We will add a map of Switzerland to Figure 1. The shading represents the hillshade and not a change in vegetation cover. We will remove the hillshade and update the figure.**

Fig. 2: please make air temperature line thicker and improve grey bars, which are not so visible

**We will improve the figure.**

Fig. 5: The snow depth subplots could be taller to increase visibility;

**The same snowpack data are already shown in Figure 2. The snowpack data in Figure 5 are only shown to indicate the timing of snow-free periods at the sites.**

Fig. 7: what is the meaning of the grey dashed line in all subplots?

**We will specify this in the figure caption: including grey dashed lines that represent the range between -85 ‰ and -75 ‰.**

Fig. 8: why are event #3 and #4 results unrealistic?

**The event water fraction resulted in negative results because the isotopic compositions of stream water did not respond to the isotopic composition of the snowpack outflow (event water). We will clarify this in the figure caption.**

**References**

Beniston, M.: Climatic change in mountain regions: A review of possible impacts, edited by H. F. Diaz, Kluwer Academic Publishers, Dordrecht., 2003.

Genereux, D.: Quantifying uncertainty in tracer-based hydrograph separations, Water Resour. Res., 34(4), 915–919, doi:10.1029/98WR00010, 1998.

Koeniger, P., Hubbart, J. A., Link, T. and Marshall, J. D.: Isotopic variation of snow cover and streamflow in response to changes in canopy structure in a snow-dominated mountain catchment, Hydrol. Process., 22(4), 557–566, doi:https://doi.org/10.1002/hyp.6967, 2008.

McCabe, G. J., Clark, M. P. and Hay, L. E.: Rain-on-snow events in the western United States, Am. Meteorol. Soc., 88(3), 319–328, doi:https://10.1175/BAMS-88-3-319, 2007.

Surfleet, C. G. and Tullos, D.: Variability in effect of climate change on rain-on-snow peak flow events in a temperate climate, J. Hydrol., 479, 24–34, doi:https://doi.org/10.1016/j.jhydrol.2012.11.021, 2013.

Zierl, B. and Bugmann, H.: Global change impacts on hydrological processes in Alpine catchments, Water Resour. Res., 41(2), 1–13, doi:10.1029/2004WR003447, 2005.

**Response to reviewer 2, Roman Juras**

**We thank Roman Juras for revising and commenting the manuscript. We have reproduced those comments below (in normal type), with our responses (in bold).**

General comments: The paper presents very interesting and current topic about snowpack out-flow contribution to the catchment outflow during rain-on-snow (ROS) events. The authors identified ten ROS events during two winter seasons, where the effect of snow cover and further snowpack outflow to the stream were analysed. Authors employed two-component hydrograph separation method using natural stable water isotopes and enhanced system of water sampling. I like the study very much, because understanding of the hydrological processes during ROS is still not sufficient and this study aims to contribute to this knowledge. It is an interesting study and worth to publish in HESS. Nevertheless, I recommend to do some minor revisions and I also have a couple of suggestions to improve the study.

My major point to the study is that the authors should present results from the hydrograph separation and provide more information about snowpack outflow composition. Since the isotopic content of rain and snowmelt during ROS events were sampled, the rainwater contribution in the outflow can be easily calculated.

**We will include an additional analysis of the rainwater contribution to snowpack outflow in the supplement. However, we want to stress that most events could not be analysed because the pre-event signature of snowpack outflow was not known (i.e., no snowpack outflow occurred before the event).**

The authors can also provide the separated hydrograph with all the components.

**We will provide the hydrograph timeseries in the revised manuscript.**

The authors define in section 2.2.1 the ROS event. Maybe I just missed something, but from this definition it seems that duration of ROS equals duration of rain. This does not match with the values in Tab. 1 (see columns Start time, End time and Rain-fall duration). This issue is also connected with total ROS outflow volumes. Please describe it clearer.

**We will clarify the definition of ROS events in the revised manuscript in section 2.1.1.**

The Introduction section usually provides in the end some basic goals of the paper. I miss this part in the particular section. Please reformulate the last paragraph (Page 3, lines 30 – 33, Page 4, lines 1 – 3).

**We will specify the research hypothesis and the goal of this paper in the introduction.**

Although, I am not a native english speaker, I recommend some proof reading regarding the language.

Specific comments:

• Please use elevation units as "m a.s.l." and not "m asl".

       **We will correct this.**

• Please present what time zone do you use (UTC, CET, etc.).

**We will specify this.**

• Figure1: Can you add an information about coordinate system of the map and how far is HG site from the catchment. You should also add a small map of Switzerland, where the study site is located.

**A map of Switzerland and information about the distance between the HG site and the MG site/catchment will be added in the revised manuscript.**

• Lot of technical information regarding the field monitoring system is provided (page 5, lines 15 – 18, page 1 – 5). It would be beneficial to better readers clarity if you present these information in tabular form. The sketch or photograph of the monitoring system would be also very practical and provide better view how the system works.

**An earlier paper:**

Rücker, A., Zappa, M., Boss, S. and von Freyberg, J.: An optimized snowmelt lysimeter system for monitoring melt rates and collecting samples for stable water isotope analysis, J. Hydrol. Hydromechanics, 67(1), 20–31, doi:10.2478/johh-2018-0007, 2019.

**provides a detailed description of the field monitoring system. We have only included this reference in the manuscript in order to shorten the methods section.**

• If you state just water stable isotopes as such, do not use $\delta$symbol, but only $_2$H and $_{18}$O (Page 3, line 3). Delta symbol refers to some defined standards.

**We will change this in the revised manuscript.**

• You mention that the snow was sampled by a snow tube (Page 7, lines 27 – 28). Do you use any standardised tube? What is the material of the tube?

**We did not use a standardized tube; the tube was custom-made at WSL from from glass fibre with epoxy resin and edges made of steel.**

• Please be consistent with presenting the time intervals. You often mix numbers and text information, like 10-minute x ten minute (i.e. Page 7, lines 7, 9).

**We will correct this.**

• Page 16, line 1: According to Fig.2 the snow depth at HG does not look 97 cm deeper than MG.

**This was a typo, we intended to present this number as a depth of the HG snowpack and not as the difference between the HG and MG snowpack depths.**

• Page 16, line 14 – 15: Do you have any isotopic signature results of the through fall? Can you compare it with rainfall on the open sites?

**During the observed winter 2016/17, we did not explicitly sample throughfall at the forested site.**

• Page 18, line 2: How did you estimate the cold content of the snow? Did you also measure the snow temperature or did you just guess it from the air temperature? If you consider just mean air temperature, how long prior to the event? Maybe you should rather use cumulative

temperature from last x hour. Nevertheless, this statement is quite tricky, because the higher cold content does not always mean that more incoming rainwater is stored in the snowpack. Water storage is more related to the snow stratigraphy and layering.

**Thank you for this thought, we will implement this into the revised manuscript.**

• Figure 4: Can you add $r^2$ values to all subplots?

**This will do this.**

• Page 20, line 4: How do you define lag times?

**We will specify lag times in the revised manuscript:**

**Lag time is the time between the beginning of the ROS event and the first response of the snowpack outflow; the first response is defined as an increase of snowpack outflow by at least 0.05 mm relative to the previous measurement.**

• Page 20, line 6: How do you estimate saturation of the snowpack?

**We did not directly measure saturation of the snowpack, however, we assume that the snowpacks at the MG and HG sites were ripe prior to event #6 because snowpack outflow volumes were continuously above 0.05 mm during the previous days. This explains why the snowpack outflow volumes immediately increased with the onset of the ROS event.**

• Table 3: What does represent the last column (Rainfall MG) of the table? There are used two terms in figure 6 –rain and rain-on-snow. Please be consistent with the naming.

**We adapt these terms in the revised manuscript.**

There are presented results of different water contribution to the catchment outflow only during peak discharge. Can you also present results of outflow composition from the entire event period?

**We will include the plots of the hydrograph separation results in the revised manuscript or supplement. In addition, we will provide the hydrograph separation results (during peak flow and during maximum contribution of snowpack outflow to streamflow) as a table in the supplement.**

**Response to reviewer 3, Daniele Penna**

**We thank Daniele Penna for revising and commenting the manuscript. We have reproduced those comments below (in normal type), with our responses (in bold).**

General comment
First of all, I apologize with the Authors and Editor for my late review.
This is a very interesting manuscript that focuses on the role played by rain-on-snow (ROS) events in enhancing snowpack outflow and thus snowmelt, ultimately contributing to stream runoff. I worked for some years in a snow-dominated catchment and I had the opportunity to observe the significant impact that ROS events have on the catchment hydrological response in the melting period. Therefore, experimental work that provides a better understanding of the controls on snowmelt contribution to streamflow during ROS events is welcome and certainly appreciated by the readers of HESS. The manuscript is well written, solidly structured, nicely illustrated, with updated and relevant references, and the data well support the results interpretation. I basically agree with the comments by the two other Reviewers and I overall like the response of the Authors. I have only a few specific comments that I hope can contribute to improving the manuscript. In the end, I recommend a minor revision before publication.

Specific comments
- In agreement with Reviewer 1, I also noticed the lack of a clear and testable research hypothesis stemming from the knowledge gaps defined earlier in the Introduction. The Authors replied that the main hypothesis ": : :is that vegetation and elevation substantially affect the generation and the isotopic composition of snowpack outflow, and thus snowmelt contribution to streamflow. In my opinion, this reply is not fully satisfactorily.

First, "vegetation" is quite a vague term in this context: reading the rest of the manuscript and knowing the area it is clear that this term refers to forest trees but, in principle, this could be valid for understory vegetation as well. So, I suggest being more specific here.

> **We will clarify in the revised manuscript that vegetation is meant to be forest canopy.**

Secondly, what does it mean that vegetation and elevation affect snowpack outflow generation? I guess the Authors mean outflow amount or volumes, but again this should be specified. Most importantly, this is only the general hypothesis. I suggest to complement it with some specific hypotheses or specific research questions that better address the core of this work and around which the Results and Discussion section could be built.
For instance, one specific research question could focus on the role of rainfall characteristics and initial snowpack properties on the variability of snowpack outflow volume.

Another specific research question could deal with the spatial variability of snowmelt contribution to streamflow in the catchment (comparison of hydrograph separation results among the three sites) and a third one to the temporal variability of snowmelt contribution to streamflow (comparison of hydrograph separation results among different ROS events). These are only suggestions but I think that structuring the Results section so that its parts reflect the specific questions posed at the beginning would tell a clearer story and accompany the reader in a more linear way.

**Thank you for this suggestion. We will propose a more general research hypothesis and include four more specific research questions at the end of the introduction.**

- I think that what would be really interesting and novel is the application of three component hydrograph separation to quantify the proportion of rainfall and pre-event snowmelt during ROS events. As far as I understood, the instrumental design and the sampling scheme would allow for the application of this mixing model that, of course, requires the availability of a second tracer. Is there any additional tracer available? Is the application feasible? Are there theoretical or practical constraints that prevent this analysis? I wonder if the Authors already planned a follow up of this study considering this aspect. A comment on this is welcome.

**We have also measured major anions and cations in all samples and it might be possible to use an additional tracer (such as magnesium, calcium) to separate the pre-event signature (high solute concentrations) from the rainwater and snowpack outflow signature (low solute concentration). We are planning on performing such an analysis, which, if it works, will result in a separate publication.**

- At lines 268 and 269 the Authors stated that the pre-event tracer signature (by the way, talking about isotopes I think that the terms "signature" and "composition" are more appropriate than "concentration") was determined by sampling the stream on the day prior the ROS event. In a previous study in a snowmelt-dominated Alpine catchment (Penna et al., 2016, JoH, https://doi.org/10.1016/j.jhydrol.2016.03.040), we compared two different methods to determine the pre-event stream signature for two component hydrograph separation during snowmelt, i.e. the average of several samples taken during baseflow and a sample taken before the snowmelt-induced runoff event. We found, in some cases, marked differences in the estimated snowmelt proportions in the stream using the two methods, and we related these differences to the fact that streamflow may have contained a small amount of residual snowmelt water at night, especially late in the melt season, so that meltwater influenced the isotopic composition of the stream between melt events. In the case presented by the Authors, the ROS events occurred in winter (Jan-March) and so this effect might not be so important but, nevertheless, I wonder if this effect could happen here as well at least in the late winter events (e.g., March). A comment on this could be useful.

**We will replace "concentration" with "composition" or "signature" in the revised manuscript.**

**For five of our six ROS events, pre-event water signatures were very similar ($\delta^{18}$O=-11.5±0.3 ‰, $\delta^2$H=-81.5±1.4 ‰), so that using a single pre-event water signature (determined from baseflow samples) would not substantially change our hydrograph separation results.**

**For event #5, pre-event water signatures were slightly lighter ($\delta^{18}$O= -12.8 ‰, $\delta^2$H= -86.3 ‰) than the average of the other five events due to very light rainfall during the preceding event #4. If the pre-event water signature would correspond to the mean value ($\delta^{18}$O=-11.5±0.3 ‰, $\delta^2$H=-81.5±1.4 ‰), we would underestimate the snowpack outflow contribution during event #5 (i.e., absolute percent differences were between 16 and 530). However, in our analysis we treat the stream water sample prior to each ROS event as our pre-event water regardless of whether another event occurred beforehand. Thus, we consider each event independently and not relative to baseflow conditions.**

Minor comments and technical corrections

L126. Remove "at mid elevations".

**We will remove this in the revised manuscript.**

L147-148. I suggest shortening the title.

**We will change this.**

L179: What is the relative measurement uncertainty of the tipping bucket?
**We gave the average measurement uncertainties in L172, so we assume that this gives the required uncertainties.**

L223. Remove the delta sign, it's not needed here.

**We will remove this in the revised manuscript.**

L255. Replace "concentration" with "composition".

**We will replace this in the revised manuscript.**

L259-260. Did the Authors/technicians apply any procedure to mitigate the carry over (memory) effect that can affect laser isotopic measurements when analysing subsequent samples with much different isotopic composition?

**Every sample was measured a minimum of 6 times.  To reduce the memory effect only the last 3 results were used and averaged to derive the isotopic composition of each individual sample.**

Fig. 3b: Add "rainfall" before the word "retention" below the 1.1 line.

**This will be added.**

L445-446. I suggest skipping this, redundant with what previously mentioned in the M&M section.

**We would intend to leave this in the revised manuscript, so the definition about snowpack water budget will be certainly clear to the reader.**

L456. Is the regression statistically significant? Can the Authors report the p-value of this regression?

**We will add the p-values of these linear regressions.**

L540-542. This sentence is not necessary and can be skipped.

**We will remove this sentence.**

Fig. 6. It is not immediately clear to me which rain samples are, which snowmelt samples, and ROS samples and bulk snow, so I suggest making the box plot clearer.

**We will do this in the revised manuscript.**

In addition, did the Authors perform a statistical analysis in order to check for the differences in isotopic composition?

**We will perform the analyses to check for differences in isotopic compositions between the different sources (at each site) and between the different sites (for the same source). We will include this information in the revised manuscript.**

Moreover, I wonder if the slope of the regression lines in the dual isotope space (Fig. 6d-f) is statistically different between the MG site and the HG and MF sites (see, for examples, another isotopic study on rain and snowmelt in the Alpine catchment mentioned above, Penna et al., 2017, HP, https://doi.org/10.1002/hyp.11050). This could be performed and discussed in the light of the inter-site comparison in rain and snowmelt isotopic composition.

**We have performed a statistical analysis and found that the slopes of the regression lines were not statistically different (p-value > 0.3). We will include this information in the revised manuscript.**

L658. Replace "concentration" with "composition".

**This will be changed in the revised manuscript.**

L706-707. Which assumptions were violated to have unrealistic results?

**The isotopic composition of the stream water and the rainwater were overlapping during event #4. We will clarify this in the revised manuscript.**

Table 1: Unpaired two samples t-Tests of the differences between the four sample types (snowmelt, rain-on-snow, rain, bulk snowpack at each site) and between the three sites (HG, MG, MF site). Upper right triangle: t-values (in italic font); lower left triangle: p-values (regular font). Statistically significant differences, i.e., *p*-values < 0.01, are shown in bold font. Grey fields indicate sample combinations that are not informative.

| Location and sample type | HG_SPO Site | | | | MG_SPO Site | | | | MF_SPO Site | | |
|---|---|---|---|---|---|---|---|---|---|---|---|
| | Snowmelt | ROS | Rain (no snowpack) | Bulk Snowpack | Snowmelt | ROS | Rain (no snowpack) | Bulk Snowpack | Snowmelt | ROS | Rain (no snowpack) |
| **HG_SPO Site — Snowmelt** | | *-1.811* | no rain | *-0.131* | *1.335* | | | | *-0.163* | | |
| **HG_SPO Site — ROS** | 0.075 | | no rain | *1.408* | | *0.987* | | | | *-1.049* | |
| **HG_SPO Site — Rain (no snowpack)** | no rain | no rain | | | | | no rain | | | | no rain |
| **HG_SPO Site — Bulk Snowpack** | 0.896 | 0.169 | no rain | | | | | *0.759* | | | |
| **MG_SPO Site — Snowmelt** | 0.185 | | | | | *-1.543* | *3.568* | *-0.091* | *1.262* | | |
| **MG_SPO Site — ROS** | | 0.334 | | | 0.128 | | *3.289* | *1.263* | | *1.469* | |
| **MG_SPO Site — Rain (no snowpack)** | | | no rain | | **<0.01** | **<0.01** | | *-2.778* | | | *-0.039* |
| **MG_SPO Site — Bulk Snowpack** | | | | 0.454 | 0.928 | 0.220 | **<0.01** | | | | |
| **MF_SPO Site — Snowmelt** | 0.871 | | | | 0.210 | | | | | *-0.328* | *2.077* |
| **MF_SPO Site — ROS** | | 0.304 | | | | 0.159 | | | 0.744 | | *1.740* |
| **MF_SPO Site — Rain (no snowpack)** | | | | | | | 0.969 | | 0.042 | 0.092 | |

**Table 2: Coefficients of slope and intercept including standard error of the three sites (HG, MG, MF site). Analysis of variance (ANOVA) of the slopes of the regression lines in the dual isotope space at the HG_SPO, MG_SPO and MF_SPO sites. Upper right triangle: t-values (in italic font); lower left triangle: p-values (regular font). The three slopes are not statistically different, i.e., *p*-values > 0.01.**

| Location | Coefficients | | Analysis of Variance | | |
|---|---|---|---|---|---|
| | Slope | Intercept [‰] | HG_SPO | MG_SPO | MF_SPO |
| HG_SPO | 8.03±0.2 | 13.7±2.6 | | *-1.602* | *-0.689* |
| MG_SPO | 7.5±0.1 | 4.5±0.8 | 0.246 | | *-1.186* |
| MF_SPO | 7.7±0.2 | 10.2±1.9 | 0.770 | 0.462 | |

[revised manuscript text omitted]
 differences in snowpack outflow generation can be transferred to the effect of forest canopy (e.g., coniferous forest).  A third snowmelt lysimeter system was installed at a grassland site at 1405 m a.s.l. higher elevation (1405 m a.s.l.m asl) site (the HG site, or high-elevation grassland site).  outside of the maintained field sites

by the WSL in the Erlenbach catchment at a location where . The HG site is approximately 2 km away from the MG site and was chosen because of technical reasons (e.g., power supply was available). ThisThe measurements from the HG site wereis usedassumed asto be a representative field site for the 1400 m elevation zone of the Erlenbach catchmenthigher elevations; however, we acknowledge that, the HG site was located on a flat, slightlywhereas its southnorth-orientedfacing plateau, whereas aspectthe does not represent the east oriented Erlenbach catchment is characterized by a sequence of flat plateaus and west-facing slopes. The terrain at the meteorological station and the MG site iswas relatively flat, whereas the MF and HG sites arewas located on slightly sloped terrain facing westsloping ridges. near the southern border of the Alptal catchment, because an installation at this altitude within the Erlenbach catchment was technically not feasible. The forest at the MF site is dominated by a forest consisting of Picea abies and Abies alba, whereas the MG and HG sites are located on grassland. The MG and the MF sites are were located 250 m apart from each other with an elevation difference of only 30 m. . The elevation, slope and aspect are similar at the MG and MF sites, so that differences in snowpack outflow generation can be transferredrelated mainly to the effect of forest canopycover (e.g., coniferous forest). 
[revised manuscript text omitted]
_{\text{spo,k}}\right)^2}\right)\text{SE}_{V_{\text{spo,k}}}\right]^2}\ . \tag{3}$$

In Eq. (3), $\text{SE}_{C_{\text{spo}}}$ is the standard error of the isotope data, which is assumed to be the measurement uncertainty of the

15  isotope analyser, and $\text{SE}_{V_{\text{spo}}}$ is the standard error of the mean of the three individual snowpack outflow volumes measured with the individual lysimeter funnels. Using Gaussian error propagation (Genereux, 1998), the uncertainty of $F_{\text{spo}}$ was estimated as

$$\text{SE}_{F_{\text{spo}}} = \sqrt{\left(\frac{-1}{(C_{\text{pe}} - C^*_{\text{spo}})}\text{SE}_{C_{\text{S}}}\right)^2 + \left(\frac{C_{\text{S}} - C^*_{\text{spo}}}{(C_{\text{pe}} - C^*_{\text{spo}})^2}\text{SE}_{C_{\text{pe}}}\right)^2 + \left(\frac{C_{\text{pe}} - C_{\text{S}}}{(C_{\text{pe}} - C^*_{\text{spo}})^2}\text{SE}_{C^*_{\text{spo}}}\right)^2}\ . \tag{4}$$

We assume that the standard errors of $C_{\text{S}}$ and $C_{\text{pe}}$ ($\text{SE}_{C_{\text{S}}}$ and $\text{SE}_{C_{\text{
[revised manuscript text omitted]
. Within this study, the contribution of rainwater to snowpack outflow was estimated indicating the retention of rainwater in the snowpack. However, it was challenging to calculate the estimates for all sites during the observed ROS events. Often a pre-event isotopic composition of the snowpack outflow could not be obtained, because of limited snowpack outflow generation prior to the ROS event. Thus, the results of this analysis can be found in the supplementary material and it is advised to not draw any general conclusion. Here we present our results based on $\delta^2H$, for which the temporal variations in stream water were larger, and the measurement uncertainties were smaller, compared to $\delta^{18}O$.

*2. IHS results are highly variable across sites and rain-on-snow (ROS) events*

Figure 8Figure 8 summarizes the estimated contributions of rainfall rainwater and snowpack outflow to peak streamflow during the six ROS events. Snowpack outflow contributions to peak streamflow varied among the six ROS events and the three snowmelt lysimeter sites, ranging from 34±7 % to 42±2 % at the HG site, from 13±1 % to 58±3 % at the MG site, and from 7±4 % to 91±20 % at the MF site (Figure 8Figure 8, Table 3Table 3). The maximum fractions of snowpack outflow during each event were different compared to the contribution to peak streamflow (Table A2 the supplement). Maximum fractions of snowpack outflow to streamflow during the events most often occurred were derived mostly for the day after the peak streamflow, and were often significantly higher than during peak streamflow (for example, using $\delta^2H$ from the HG site as the snowmelt source during event #6, estimated snowmelt contributions to streamflow were and were up to 41±11 % higher (e.g., event #6 at HG site 34 ± 7 % during peak streamflow but 75 ± 18 % the day afterwardbased on $\delta^2H$) compared to the estimated contributions during peak streamflow (e.g., event #6 at HG site 75 ± 18 % based on $\delta^2H$). ;
[revised manuscript text omitted]
 S1: Measurements of hourly air temperature (a), daily precipitation (snow-and rainfall) and snow depth (b), and daily snowpack outflow volumes (c), measured at the mid-elevation forest site (MF) for the study period 1 November 2017 – 6 April 2018.  Panel (d) shows daily discharge at the Erlenbach catchment outlet (on log scale).  Vertical grey bars indicate the four rain-on-snow (ROS #7-#10) events during winter 2018 that are analysed in this study only at the MF site.

- Table S1: Estimated contributions of rainfall or snowpack outflow to streamflow during peak flow based on two-component isotope hydrograph separation using $\delta^{18}$O (HG: high-elevation grassland site; MG: mid-elevation grassland site; MF: mid-elevation forest site).

- Table S2: Maximum contributions of rainwater or snowpack outflow to streamflow during daily discharge based on two-component isotope hydrograph separation using $\delta^{18}$O and $\delta^2$H (HG: high-elevation grassland site; MG: mid-elevation grassland site; MF: mid-elevation forest site).  The grey boxes with bold numbers indicate a different fraction compared to the fractions during peak flow (see Table 3 in the main text).

-

  -

- Table S3: Relative contribution of rainwater to snowpack outflow during peak daily snowpack outflow based on two-component isotope hydrograph separation using $\delta^{18}$O or $\delta^2$H (HG: high-elevation grassland site; MG: mid-elevation grassland site; MF: mid-elevation forest site).

- Figure S2: Isotope hydrograph separations (IHS) using $\delta^2$H for the six ROS events during winter 2017, for "new water" end members comprised of snowpack outflow at the lysimeter sites HG (red), MG

(yellow) and MF (green) as well as for rainwater sampled at the MG site. The coloured bars indicate the rate of snowpack outflow or precipitation. The black lines indicate daily stream discharge, and the grey lines indicate the "new water" contribution from snowpack outflow or precipitation, as estimated by isotope hydrograph separation. Figure S2: δ²H-based hydrograph separation results for the six ROS events during winter 2017. Each row represents one event and each column represents the results obtained for the isotope measurements in snowpack outflow at the snowpack lysimeter sites HG (red), MG (yellow) and MF (green) as well as for for the isotope measurements in rainwater at the XXX site (e.g., MG site).

[Figure]

**Figure S1: Measurements of hourly air temperature (a), daily precipitation (snow-and rainfall) and snow depth (b), and daily snowpack outflow volumes (c), measured at the mid-elevation forest site (MF) for the study period 1 November 2017 – 6 April 2018.  Panel (d) shows daily discharge at the Erlenbach catchment outlet (on log scale). Vertical grey bars indicate the four rain-on-snow (ROS #7-#10) events during winter 2018 that are analysed only at the MF site (no measurements of snowpack outflow were available for the HG and MG site).**

**Table S1: Estimated contributions of rainfall or snowpack outflow to streamflow during peak flow based on two-component isotope hydrograph separation using $\delta^{18}O$ (HG: high-elevation grassland site; MG: mid-elevation grassland site; MF: mid-elevation forest site).**

| ROS event number |  Relative contribution to peak  daily  discharge ±SE (%) | | | |
|---|---|---|---|---|
| | Snowpack outflow HG | Snowpack outflow MG | Snowpack outflow MF |  Rainwater (Outlet) |
| #1 | a) | 117 ± 21 | 191 ± 118 | 68 ± 11 |
| #2 | 43 ±  | b) | 167± 92 | b) |
| #3 | -19 ± 15 | -59 ± 61 | 26 ± 19 | 11 ±  |
| #4 | a) | 30 ± 12 | 51 ± 16 | 205 ± 947 |
| #5 | 170 ± 061 | 41 ±  | 30 ± 06 | 2 ± 04 |
| #6 | 78 ± 26 | 12 ±  | 22 ± 08 | 09 ± 04 |

a) no snowpack outflow occurred

b) data gap

**Table S2: Maximum contributions of rainwater or snowpack outflow to streamflow during daily discharge based on two-component isotope hydrograph separation using $\delta^{18}O$ and $\delta^2H$ (HG: high-elevation grassland site; MG: mid-elevation grassland site; MF: mid-elevation forest site). The grey boxes with bold numbers indicate a different fraction compared to the fractions during peak flow (see Table 3 in the main text).**

 Numbers in bold and in  section 3.2.3

| ROS event number | Maximum relative contribution to mean daily discharge ±SE (%) | | | | | | | |
|---|---|---|---|---|---|---|---|---|
| | Snowpack outflow HG | | Snowpack outflow MG | | Snowpack outflow MF | | Rainwater (Outlet) | |
| | $\delta^{18}O$ | $\delta^2H$ | $\delta^{18}O$ | $\delta^2H$ | $\delta^{18}O$ | $\delta^2H$ | $\delta^{18}O$ | $\delta^2H$ |
| #1 | a) | a) | 1.17 ± 0.21 | 0.58 ± 0.03 | 1.91 ± 1.18 | 0.76 ± 0.30 | 0.68 ± 0.11 | 0.34 ± 0.02 |
| #2 | **0.50 ± 0.12** | **0.50 ± 0.03** | b) | b) | 1.67 ± 0.92 | 0.91 ± 0.20 | b) | b) |
| #3 | **0.09 ± 0.09** | **0.03 ± 0.04** | **0.12 ± 0.09** | **0.03 ± 0.03** | 0.26 ± 0.19 | 0.07 ± 0.04 | 0.11 ± 0.08 | 0.05 ± 0.02 |
| #4 | a) | a) | 0.30 ± 0.12 | 0.29 ± 0.04 | 0.51 ± 0.16 | 0.46 ± 0.06 | 2.05 ± 9.47 | **0.22 ± 15.51** |
| #5 | **0.18 ± 0.25** | 2.63 ± 0.64 | **0.46 ± 0.07** | **0.64 ± 0.07** | **0.35 ± 0.04** | **0.42 ± 0.01** | **0.24 ± 0.32** | **0.32 ± 0.01** |
| #6 | 0.78 ± 0.26 | **0.75 ± 0.18** | **0.33 ± 0.84** | **0.33 ± 39.78** | **0.53 ± 0.37** | **0.45 ± 13.55** | **0.23 ± 1.04** | **0.26 ± 0.04** |

a) no snowpack outflow occurred

b) data gap

*Contribution of rainwater to snowpack outflow*

Isotope hydrograph separation at the scale of an individual snowpack can potentially quantify how much rainwater contributes to snowpack outflow during a ROS event, compared to pre-event water which was already stored in the snowpack (e.g., snowmelt). Such results can thus indicate how much rainwater was retained in the snowpack, especially when a snowpack was not yet saturated with pre-event liquid water. Unfortunately we could not perform these snowpack-scale hydrograph separations whenever the pre-event isotopic composition of snowpack outflow could not be obtained due to limited snowpack outflow generation prior to the event (event #1: HG, MG and MF; event#2: HG, event #6: MF was already snow-free) or during a data gap in the rainwater sampling (event #2). Thus, the results of this analysis should be used with caution.

In some cases, the estimated contributions of rain to snowpack outflow were unrealistic (e.g., negative contribution based on $\delta^{18}O$ and/or $\delta^2H$) because the isotopic composition of snowpack outflow did not respond to that of the incoming rainwater. These results indicate that rainwater infiltrated into the snowpack and pushed out pre-event liquid water, which thus made up most of the snowpack outflow with very little contribution from current rainfall (event #3: MG; event #4: MG, MF; event #5: HG). During event #5, the relative contribution of rainwater to snowpack outflow was heterogeneous among the three snowmelt lysimeter sites. At the MF site, the snowpack was already shallow (e.g., 8.6 cm), so that rainwater contributed significantly to snowpack outflow ($74 \pm 3$ % based on $\delta^2H$) whereas the snowpack outflow at the HG site was less dominated by rainwater due to a

deeper snowpack and higher contribution of pre-event liquid water ($16 \pm 2$ % based on $\delta^2$H).  At the MG site, snowpack outflow was a mixture of both rainwater and pre-event liquid water ($49 \pm 3$ % based on $\delta^2$H).  During event #6, the contribution of rainwater to the snowpack outflow at the MG site was high (e.g., $88 \pm 1$ % based on $\delta^2$H) indicating that rainwater dominated the snowpack outflow.  This result agrees with the observations in section 3.1.3, because this rainfall (66.9 mm) caused the melt-out of the ripe and shallow snowpack (e.g., 17 cm), so that rainwater primarily contributed to the snowpack outflow.  At the HG site, the measured snowpack outflow volumes indicated that the snowpack was not yet saturated (section 3.1.2), so that more rainwater was retained in the snowpack, pushing out pre-event liquid water and leading to a small contribution of rainwater to snowpack outflow ($24 \pm 8$ % based on $\delta^2$H).

Table A 3: Contribution of rainwater to snowpack outflow during peak snowpack outflow based on two-component isotope hydrograph separation using on $\delta^{18}$O and $\delta^2$H (HG: high-elevation grassland site; MG: mid-elevation grassland site; MF: mid-elevation forest site).

Table S3: Relative cContribution of rainwater to snowpack outflow during peak daily snowpack outflow based on two-component isotope hydrograph separation withusing using on $\delta^{18}$O andor $\delta^2$H (HG: high-elevation grassland site; MG: mid-elevation grassland site; MF: mid-elevation forest site).

| ROS event number | Fraction Relative contribution to peak of daily peak dischargesnowpack outflow ±SE (%-) | | | | | |
|---|---|---|---|---|---|---|
| | Snowpack outflow HG | | Snowpack outflow MG | | Snowpack outflow MF | |
| | $\delta^{18}$O | $\delta^2$H | $\delta^{18}$O | $\delta^2$H | $\delta^{18}$O | $\delta^2$H |
| #1 | a) | a) | a) | a) | a) | a) |
| #2 | b) | b) | b) | b) | b) | b) |
| #3 | a) | a) | -0.29 ± 0.49 c) | -0.10 ± 0.22 | 0.43 ± 0.14 | 0.57 ± 0.15 |
| #4 | a) | a) | -1.97 ± 1.95 c) | -3.52 ± 2.99 c) | -1.58 ± 0.51 c) | -4.19 ± 4.38 c) |
| #5 | -0.08 ± 0.03 c) | 0.16 ± 0.02 | 0.43 ± 0.04 | 0.49 ± 0.03 | 0.54 ± 0.05 | 0.74 ± 0.03 |
| #6 | 0.26 ± 0.10 | 0.24 ± 0.08 | 0.80 ± 0.01 | 0.88 ± 0.01 | snow-free | snow-free |

a) no pre-event snowpack outflow occurred
b) data gap
c) unrealistic

b) data gap

[Figure]

**Figure S2: Isotope hydrograph separations (IHS) using δ²H for the six ROS events during winter 2017, for "new water" end members comprised of snowpack outflow at the lysimeter sites HG (red), MG (yellow) and MF (green) as well as for rainwater sampled at the catchment outlet. The coloured bars indicate the rate of snowpack outflow or precipitation. The black lines indicate daily stream discharge, and the grey lines indicate the "new water" contribution from snowpack outflow or precipitation, as estimated by isotope hydrograph separation.**

Figure A 2:

---

## Author Response (AR2)

**Response to the editor, Bettina Schaefli**

Dear Bettina,

**thank you for your comments. We have reproduced those comments below (in normal type), with our responses (in bold) including the extraction from the manuscript with information about the page and line number.**

- Intro: the references for the rain-to-snow transition zone in Switzerland are most likely not appropriate; two references (McCabe, Surflet) are not from Europe I think. Zierl & Bugmann do most likely not talk explicitly about this, neither does, to the best of my knowledge, Beniston, 2003. It is, in fact, pretty hard to find a reference for this transition zone. You can cite the hydrologic atlas (the plate on regimes) for the transition from nival to pluvial regimes; we tried very hard to find a reference, but could not find one so far. It would not be a good thing to introduce the above references as references for Switzerland.

> **Thank you for this comment. We generalized this statement and removed/added another reference p.2, L: 6:**
>
> In Switzerland, mean air temperature is predicted to increase by up to 1.6 °C by 2050 (Swiss Academies of Sciences, 2016), and in mountain regions including the Alps, climate warming is expected to make rain-on-snow events more prevalent in the future, and raise the elevation of the rain-to snow transition zone (McCabe et al., 2007; Surfleet and Tullos, 2013; Ye et al., 2008).

- HESS does not like "double letter " variable names in equations; you should not have "SE" for standard deviation in the equation, only in the text. The correct solution is to have sigma (greek) in the equations

> **We cannot use sigma due to the risk of confusing notation (sigma is used for many other equations); the notation SE should be standard in Hess; see the publication by von Freyberg et. al. 2017. However, we changed the notation of SE of the equations in section 2.5; for example (p. 11, L.16):**
>
> $\mathrm{SE}_{C^*_{\mathrm{spo}}}$ is now changed to $\mathrm{SE}(C^*_{\mathrm{spo},i})$

- I did not understand what the index j is in the equations. In fact, in eq. 2 it is not clear what you mean with "since the beginning" (of what?); it would help to say what j is explicitly

> **We clarified this, see p. 10, L. 12:**
>
> The isotopic composition of snowpack outflow at day *i* was calculated as the incremental volume-weighted mean using the measured volumes of snowpack outflow or rainfall since the beginning of the event on day *j* (McDonnell et al., 1990):

-where does equation 6 come from? I do not think it is a well-known equation; you do not say what CR star is. I can deduce of course, but it could be stated explicitly

> **We clarified this, see p. 11, L. 13:**

The standard error of the volume-weighted rainwater isotopic composition, $\mathrm{SE}(C_\mathrm{R}^*)$, was estimated analogously to Eq. (5) of von Freyberg et al. (2017):

- section 3.1.2: you say that the snowpacks at MF site were 5cm and 8cm. You say, however that ROS events are identified as such only if snowpack>10cm

**We clarified this, see p. 17, L. 10:**

Note that for events #3 and #4 the MF site had snowpacks of only 5 cm and 8 cm, respectively (although these were still identified as ROS events because snowpacks at the reference site, MG, were 29 and 43 cm, respectively).

- you have the term "catchment recharge" somewhere; what is this?

**We removed this term as it does not give any important information. See p. 25. L. 27:**

Similar to other studies (Gustafson et al., 2010; Taylor et al., 2002a), our data show that snowpack outflow generation can be much more variable in time than it would be implied by weekly bulk snow samples alone.

- the conclusion starts with saying that ROS events are predicted to increase in Switzerland. I do not think that we have predictions on this for Switzerland. Could you either add a reference or remove? otherwise your paper might be cited in the future for this statement.

**We generalized this statement in the revised manuscript; see p.32, L.9:**

In many mountain regions, global warming is predicted to lead to more frequent rain-on-snow (ROS) events, which can enhance snowmelt and increase the risk of destructive winter floods.

**References**

von Freyberg, J., Studer, B. and Kirchner, J. W.: A lab in the field: High-frequency analysis of water quality and stable isotopes in stream water and precipitation, Hydrol. Earth Syst. Sci., 21(3), 1721–1739, doi:10.5194/hess-21-1721-2017, 2017.

**Minor Revision**

**Editor Decision: Publish subject to technical corrections** (22 May 2019) by Bettina Schaefli
Comments to the Author:
Dear authors

thanks for the submission of the revised version, which carefully addressed the minor comments of all reviewers. The manuscript is almost ready for publication, upon correction of some technicalities.

Non-public comments to the Author:
Dear Andrea

I went carefully throught the manuscript and have a few final comments that should/could be addressed before publication (without further review from the editor side):

- Intro: the references for the rain-to-snow transition zone in Switzerland are most likely not appropriate; two references (McCabe, Surflet) are not from Europe I think. Zierl & Bugmann do most likely not talk explicitely about this, neither does, to the best of my knowledge, Beniston, 2003. It is, in fact, pretty hard to find a reference for this transition zone. You can cite the hydrologic atlas (the plate on regimes) for the transition from nival to pluvial regimes; we tried very hard to find a reference, but could not find one so far. It would not be a good thing to introduce the above references as references for Switzerland.
- HESS does not like "double letter " variable names in equations; you should not have "SE" for standard deviation in the equation, only in the text. The correct solution is to have sigma (greek) in the equations
- I did not understand what the index j is in the equations. In fact, in eq. 2 it is not clear what you mean with "since the beginning" (of what?); it would help to say what j is explicitely
-where does equation 6 come from? I do not think it is a well-known equation; you do not say what CR star is. I can deduce of course, but it could be stated explicitly

- section 3.1.2: you say that the snowpacks at MF site were 5cm and 8cm. You say, however that ROS events are identified as such only if snowpack>10cm

- you have the term "catchment recharge" somewhere; what is this?

- the conclusion starts with saying that ROS events are predicted to increase in Switzerland. I do not think that we have predictions on this for Switzerland. Could you either add a reference or remove? otherwise your paper might be cited in the future for this statement.

**Uploaded Files validated** (15 May 2019) by Anna Wenzel

**File Upload** (14 May 2019) by Andrea Rücker ▸ Manuscript ▸ Supplement ▸ Author's Response ▸ Abstract